# Building regulatory landscapes reveals that an enhancer can recruit cohesin to create contact domains, engage CTCF sites and activate distant genes

Niels J. Rinzema, Konstantinos Sofiadis, Sjoerd J. D. Tjalsma, Marjon J. A. M. Verstegen,
Yuva Oz, Christian Valdes-Quezada, Anna-Karina Felder, Teodora Filipovska, Stefan van der Elst,
Zaria de Andrade dos Ramos, Ruiqi Han, Peter H. L. Krijger ⬤ and Wouter de Laat ⬤ ✉

**Developmental gene expression is often controlled by distal regulatory DNA elements called enhancers. Distant enhancer action is restricted to structural chromosomal domains that are flanked by CTCF-associated boundaries and formed through cohesin chromatin loop extrusion. To better understand how enhancers, genes and CTCF boundaries together form structural domains and control expression, we used a bottom-up approach, building series of active regulatory landscapes in inactive chromatin. We demonstrate here that gene transcription levels and activity over time reduce with increased enhancer distance. The enhancer recruits cohesin to stimulate domain formation and engage flanking CTCF sites in loop formation. It requires cohesin exclusively for the activation of distant genes, not of proximal genes, with nearby CTCF boundaries supporting efficient long-range enhancer action. Our work supports a dual activity model for enhancers: its classic role of stimulating transcription initiation and elongation from target gene promoters and a role of recruiting cohesin for the creation of chromosomal domains, the engagement of CTCF sites in chromatin looping and the activation of distal target genes.**

Tissue-specific and developmentally restricted expression of genes is often controlled by enhancers, regulatory DNA elements that can act over distance to activate gene expression. In the mammalian genome, enhancers may be located near their target gene, but they can be separated over hundreds of kilobases. Chromatin looping is thought to enable distal enhancers to approach and activate target genes. Indeed, individually studied developmental genes have been found to form preferred contacts with distal enhancers for their activation[1], as measured by chromosome conformation capture (3C) methods[2,3]. Genome-wide, a general correlation between enhancer-promoter (E-P) contact frequencies and transcriptional activity has been uncovered by these methods[4–7]. Also, non-coding disease-associated genetic variants, identified by genome-wide association studies (GWASs), could be successfully linked to target genes when considering their 3C-based chromatin contacts[8]. As an enhancer has been found to activate expression when it was forced to loop to a distant gene, E-P looping seems a requirement for, more than a consequence of, long-range gene activation[9].

Cohesin is responsible for chromatin loop formation in interphase chromosomes. In vitro, the ring-shaped cohesin complex can extrude DNA to form loops[10,11], exactly as it has been predicted to create chromatin loops in vivo[12,13]. DNA-bound CTCF is the most dominant factor that binds and halts the DNA-extruding cohesin complex on chromatin and to anchor DNA loops in mammalian cells[14–18]. CTCF-anchored loops are the most readily detectable loops in genome-wide Hi-C contact maps[16], highlighting their frequent recurrence across cells. They span sub-megabase domains, also known as topologically associating domains (TADs), containing

sequences that preferentially self-interact, and they form boundaries that hamper contacts between neighboring domains.

Without cohesin, both CTCF loops and contact domains dissolve in Hi-C chromatin contact maps, but overall steady-state gene expression levels seem remarkably stable[19,20]. This raised doubts about whether chromatin looping was really necessary for enhancers to control gene expression over distance, doubts that were further fueled by live-cell microscopy studies that have sometimes[21], but not always[22,23], found enhancers in closer proximity to the genes that they activated. Yet other studies did observe that cohesin depletion caused loss of E-P contacts[24,25] and reduced enhancer-dependent expression[24–27] of many, but not all, genes, suggesting that both cohesin-dependent and cohesin-independent mechanisms exist for long-range gene regulation[25,28]. Acute depletion of WAPL, the factor that releases cohesin from chromatin, caused repositioning of cohesin from tissue-specific enhancers to CTCF boundaries, disrupted E-P looping and downregulated expression of tissue-specific genes controlled by such enhancers[24]. This suggests that these enhancers can serve as cohesin entry sites and that active cohesin loading is required for their productive interaction with target genes[24]. Support for this model came from findings that NIPBL, the cohesin loading factor, is preferentially associated with enhancers[29,30] and appears to be required for long-range gene regulation[20]. Looping between CTCF sites, which depends on cohesin, has been found to correlate with the presence of active promoters and enhancers in the intervening chromatin[31,32], further suggesting that active regulatory DNA elements can act as entry sites for the loop-extruding cohesin machinery.

CTCF boundaries insulate chromatin domains not only physically but also functionally, as they can obstruct enhancers to activate

Oncode Institute, Hubrecht Institute-KNAW and University Medical Center Utrecht, Utrecht, the Netherlands. ✉e-mail: w.laat@hubrecht.eu

genes in neighboring domains. Consequently, cognate enhancers of a given gene are typically found in the same contact domain[15,33–36]. Within a contact domain, a proximal gene in principle has an advantage over a distal gene for activation by a shared enhancer[36–38]. Promoter mutations that interfere with enhancer contacts can re-direct the enhancer to contact and activate more distal genes[39,40].

Collectively, this suggests an intricate interplay between enhancers, CTCF sites and cohesin to form contact domains and control the expression of distant genes. To experimentally investigate this further, we took a bottom-up approach and built a large series of different regulatory landscapes in an inactive chromatin environment. In this chromatin setting, we find that a developmental enhancer can recruit the cohesin loop-extrusion machinery to promote longer-range chromatin contacts, build contact domains and enable long-range gene activation. Since the same E-P pair relies on cohesin for gene activation strongly when separated over large distances (>100 kb), mildly when separated over 47 kb and not at all in proximal (<11 kb) configurations, our data show that linear distance is important in determining whether an enhancer requires cohesin for 'long-range' gene regulation.

## Results

**Increased E-P distances lower expression level and stability.** To investigate the role of regulatory DNA sequences in building transcriptional regulatory landscapes and forming topological domains, we experimentally searched for a suitable chromatin environment. This, we reasoned, had to be a transcriptional neutral or repressive chromosomal segment. We used the 6.5-kb human β-globin micro-LCR[41] (μLCR), a prototype of a strong tissue-specific enhancer, as a regulatory DNA sequence, and a β-globin gene (*HBG1*) promoter-driven green fluorescetn protein (GFP) reporter as its target gene (Fig. 1a). We randomly integrated them as a single construct in the genome of erythroleukemia K562 cells and selected high-GFP-expressing clones. We then used Cre recombinase to remove the μLCR in order to find clones whose high reporter-gene expression strictly relied on the μLCR. These clones, we reasoned, had integrated the transgene in a transcriptional non-supportive chromatin environment. The integration sites were mapped and, with help of publicly available Hi-C and chromatin immunoprecipitation and sequencing (ChIP–seq) data sets, we selected an integration site on chromosome 18 (Chr18: 19609009) inside a relatively large (nearly 600 kb) and diffuse structural chromatin domain that was covered with repressive histone H3 trimethylated at K27 (H3K27me3) marks, had few small H3 acetylated at K27 (H3K27Ac) sites and lacked expressed genes. The right boundary of this domain, 500 kb away from the integrated reporter gene, showed a striped pattern in Hi-C, indicative of anchored loop extrusion activity[30]. Elsewhere in the locus, a few selected CTCF sites showed cohesin association, implying that this locus was not devoid of natural cohesin association and activity (Fig. 1b).

We then studied the impact of E-P distance on transcriptional output. For this, we used CRISPR–Cas9 to re-insert the μLCR at 0, 11, 47 and 100 kb upstream of the reporter gene. Bulk analysis of GFP-expressing cells demonstrated that transcriptional output decreased with increasing E-P distance (Fig. 1c). K562 cells can be stimulated by hemin to further differentiate toward the erythroid lineage[42] and to upregulate LCR-mediated globin gene expression[43]. At all E-P distances, hemin treatment resulted in a roughly three- to fivefold further upregulation of the reporter gene. This showed that the inverse relation between linear E-P distance and transcriptional output was maintained under conditions that stimulated enhancer communication (Fig. 1c).

Expansion of GFP-positive sorted cells revealed that some cells lost their ability to express GFP over time, even if they were stimulated by hemin. To further investigate this, we again site-specifically inserted the μLCR at the selected distances and bulk-sorted cell populations for GFP expression for 2 consecutive weeks. We then FACS monitored them weekly, each time after a 2-day hemin induction. When immediately flanking the reporter gene (at 0-kb distance), the enhancer conferred long-term stable expression: even after 5 weeks of culturing, nearly all cells (>95%) showed very high GFP expression that was enhancer-dependent. With the enhancer integerated at increasing distances from the promoter (11 kb, 47 kb, 100 kb, respectively generating clonal cell lines E11, E47 and E100), however, the enhancer correspondingly lost the capacity to protect the reporter gene from silencing (~20% (E11), 30% (E47) and 50% (E100) silenced cells after 5 weeks) (Fig. 1d and Extended Data Fig. 1a). At all distances, GFP protein levels in the long-term active cell population remained as high, or nearly as high, as before (Extended Data Fig. 1a). The linear distance between the enhancer and promoter in a repressive chromatin environment, therefore, is related inversely not only to transcriptional activity, but also to transcriptional stability.

Given the high local levels of H3K27me3 in this locus (Fig. 1b), we asked whether silencing was accompanied by the accumulation of this repressive histone mark at the reporter gene promoter. For this, we performed ChIP on long-term silenced and active E100 cells, and on cells lacking the enhancer (noE cells). In silenced cells, as compared with active cells, we noticed increased levels of H3K27me3 at the gene promoter. H3K27me3 levels at the reporter gene were even higher in noE cells (Extended Data Fig. 1b). Endogenous copies of the β-globin LCR prohibited such analysis of the integrated μLCR (but see below). As H3K27me3 is deposited by the PRC2 polycomb complex, this suggested that the enhancer helped counteract polycomb-mediated reporter gene silencing.

**E-P contacts do not strictly dictate transcription activity.** Given that chromatin contact frequencies generally decay exponentially with increased chromosomal distance, our data, similar to other work[36], suggested a relationship between enhancer-mediated gene activation and E-P contact frequencies. We used 4C-seq[44] to more directly analyze E-P contact frequencies. 4C-seq involves quantifying the competitive ligation events between a genomic site of interest and its spatially most proximal DNA fragments inside each cell nucleus, which then provides a semi-quantitative measure for contact frequencies in a cell population. We first selected for each E-P distance pair a clonal cell line (the enhancer lines, or E-lines), which we genetically confirmed and validated to display representative transcriptional activity and stability (Fig. 1e,g). We failed to generate bulk (multi-clonal) populations with the enhancer at 200, 300 or 400 kb (even if we co-integrated a CTCF boundary, see below), but sorted out an individual E-line expressing GFP under the control of the μLCR at 407 kb. This E407 cell line expressed GFP at lower levels (see also Extended Data Fig. 1c, for RNA quantification by reverse transcriptase qPCR) and was more prone to silencing than clones with a more proximal enhancer (Fig. 1e,g and Extended Data Fig. 1d), further showing that transcriptional activity and stability decreased with enhancer distance. When we deleted the >400-kb intervening sequence in long-term silenced E407 cells (that is, those cultured for more than 6 weeks, and weekly selected for the absence of GFP expression), rare cells with very high GFP expression were identified and clonally expanded (Fig. 1f). Genotyping confirmed that they carried the deletion that placed the enhancer at 0-kb distance from the reporter gene. Although we found that not all cells with the deletion re-expressed the reporter gene at high levels, this further supported that enhancer distance can impact expression levels and suggested that gene silencing can be reversed when moving the enhancer closer to the gene, as has been previously observed through forced chromatin looping[9].

We applied 4C-seq to the reporter gene to analyze its chromatin contacts in the different E-lines. As a proxy for E-P contact frequencies, we quantified the contacts with the μLCR and expressed this

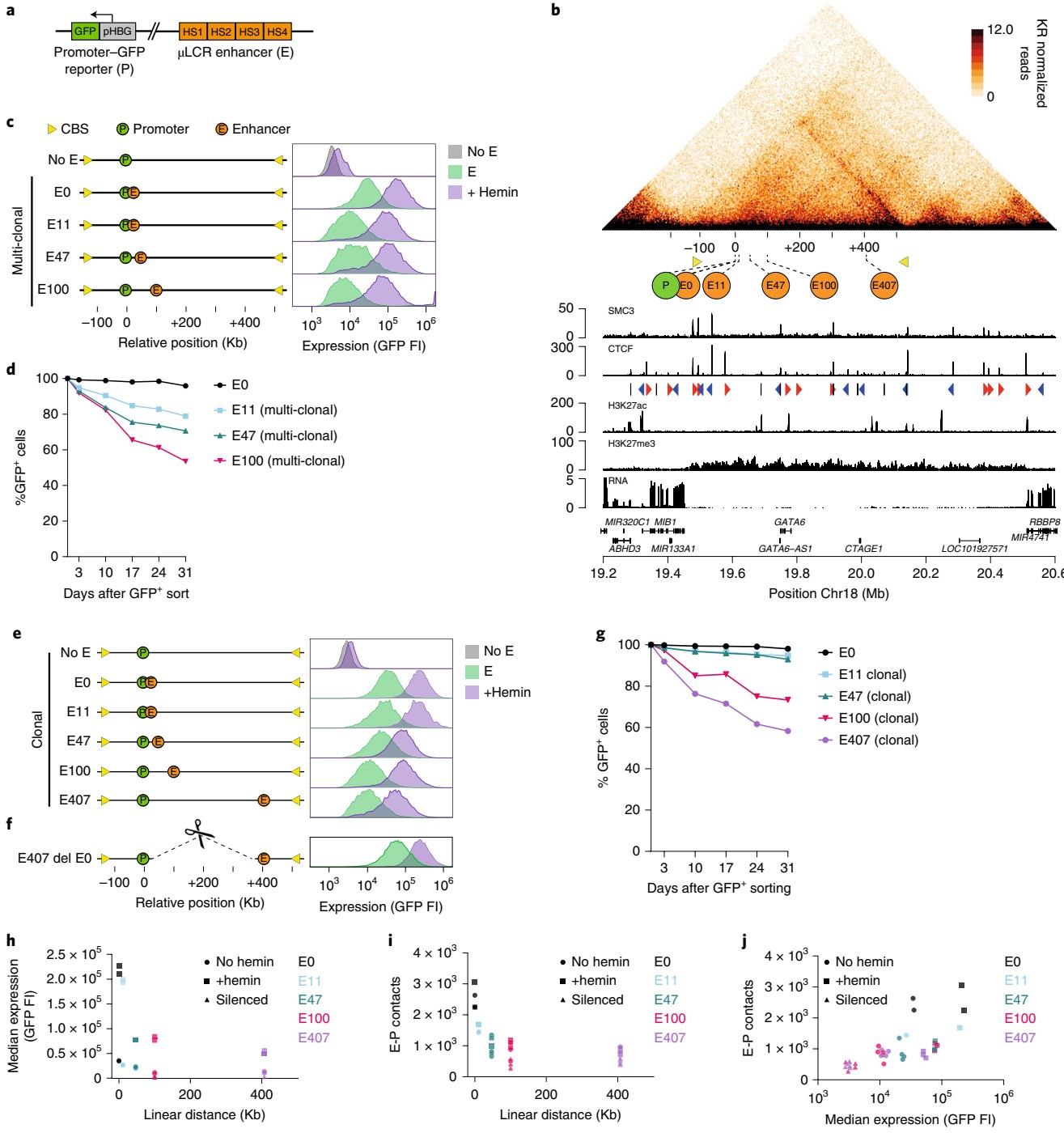

**Fig. 1 | E-P distances and contact frequencies versus gene expression. a**, Schematic representation of the reporter gene consisting of the *HBG1* promoter (P) driving GFP expression and the enhancer (E) comprising of four of the DNase I hypersensitivity sites (HS1–4) of the human β-globin locus control region (μLCR)[41]. **b**, Genomic context of integrated transgene in K562 cells: Knight-Ruiz (KR) normalized HiC contact map at 5-Kb resolution taken from ref. [16], with the different enhancer integration sites (E) indicated. For reference, yellow triangles demarcate the 'left boundary' (see text) and right domain boundary. Tracks below show: ChIP–seq tracks[61] for SMC3; CTCF binding sites (CBS), forwardly and reversely orientated CBS in red and blue triangles, respectively; H3K27ac; H3K27me3; RNA-seq; and Ref-seq genes. Genomic positions on chromosome 18 are in Mb. **c**, GFP fluorescence intensity in arbitrary units (FI) of multi-clonal (bulk) cell populations carrying no enhancer (no E) or the μLCR enhancer at 0 kb, 11 kb, 47 kb or 100 kb (E0, E11, E47, E100, respectively) upstream of the GFP-reporter, without hemin (green) or with hemin (purple) in the culture medium. **d**, Percentage of remaining GFP-positive cells in multi-clonal (bulk) cell populations carrying the enhancer at the indicated distances, after long-term cell culturing. At each time point, cells were first treated with hemin for 2 days prior to FACS analysis. *n* = 1 replicate per time point per clone. **e**, Same as in **c**, but for selected clonal cell lines carrying the μLCR enhancer at indicated distances. **f**, GFP fluorescence intensity of an E407 cell line, after deleting intervening sequences to have the μLCR enhancer placed at 0 kb from the reporter gene. **g**. Same as in **d**, but for selected E-lines carrying the μLCR enhancer at indicated distances. *n* = 1 replicate per time point per clone. **h–j**, Median reporter gene expression (GFP fluorescence) plotted against linear E-P distance (**h**), and mean 4C-seq-measured E-P contacts (mean normalized (per 1 million *cis*-reads) 4C-seq signal at enhancer) plotted against E-P linear distance (**i**) or against median GFP fluorescence (**j**). Cells were untreated (circles), treated with hemin (squares) or long-term silenced (triangles). *n* = at least 2 technical replicates/clone.

as a percentage of intra-chromosomal 4C contacts. When this was plotted against the average GFP expression levels, we observed an overall positive, but not linear, relationship between E-P contact frequencies and transcriptional activity (Fig. 1h–j). To further investigate this relationship, we took advantage of our system's ability to modulate expression levels at a given enhancer position. First, we performed similar 4C-based measurements after hemin treatment, which induced reporter gene expression in all E-lines by three- to fivefold. This upregulation was generally accompanied by a slight increase in E-P contact frequencies (Fig. 1h–j and Extended Data Fig. 2a). We then investigated E-P contacts in long-term silenced clones (see above) that no longer expressed the reporter gene, even after hemin induction. Here, E-P contact frequencies were slightly reduced as compared with those in active cells, but the now inactive gene continued to show contacts throughout the domain that could be explained only by enhancer activity (Fig. 1h–j and Extended Data Fig. 2b,c) (see also below). Collectively these results support other recent data that the relationship between gene activity and E-P contact frequencies is non-linear and that subtle changes in contact frequencies can lead to large changes in expression[36,45,46].

**Enhancer forms local self-interacting chromatin domains.** We then asked whether the integration of the reporter gene or enhancer had an impact on the topology of the locus. To test this for the reporter gene, we assayed the chromatin contacts of the integrated transgene promoter in cells lacking a co-integrated enhancer and compared these with contacts made by the endogenous sequence immediately flanking its integration site in wild-type (WT) cells. Contact profiles were almost identical (Fig. 2a and Extended Data Fig. 3), which suggested that insertion of the reporter gene itself had little impact on chromatin topology.

In contrast, when we plotted the reporter gene's 4C contact profiles in the E-lines as overlays over its contacts when it had no integrated μLCR in *cis*, it became obvious that the distantly located enhancer stimulated the gene to engage bi-directionally in longer-range contacts. Gene contacts were stimulated not only with the enhancer itself, but also with intervening and surrounding sequences, across a defined genomic interval (Fig. 2b). This genomic interval of enhancer-induced gene contacts appeared similar between all lines that had the enhancer integrated within 100 kb, and corresponded to the domain that each of these integrated enhancers preferentially contacted themselves (Fig. 2c and Extended Data Fig. 3). A different enhancer-activated contact domain was observed in the E407 E-line. Here, the ultra-far upstream enhancer exclusively stimulated contacts of the reporter gene with upstream sequences, across 400 kb of sequences toward and with the upstream enhancer (Fig. 2d,e). The E407 enhancer itself probed this same region, with clear interactions not only with the distal region containing the gene, but also with two unexpectedly prominent more specific contacts, one with an undefined intervening DNA element and the other with the flanking right CTCF-boundary (Extended Data Fig. 3). Collectively, the data show that an enhancer at different locations can stimulate a gene to engage in different chromatin contacts and induce the formation of different local contact domains.

**Smaller domains support enhancer action.** We wished to study the relationship between domain sizes and gene regulation in more detail, and introduced a 3×-hCTCF cassette, with three strong CTCF-binding sites (CBS) selected from the human genome. We successfully integrated the μLCR together with the 3×-hCTCF cassette at 0, 11, 47 and 100 kb from the reporter gene, with the CBSs oriented convergently, facing the enhancer and (distal) reporter gene (we failed to find integrations at 407 kb). We generated bulk GFP-positive cell populations, as described before, and found that, at all four integration sites, the presence of flanking CBSs resulted in higher levels of transgene expression, even after hemin induction

(Fig. 3a). Furthermore, we found that the flanking CBSs also helped the enhancer at all distances protect the transgene from silencing (Fig. 3b and Extended Data Fig. 1e). We then selected representative clonal cell lines with the enhancer-3×-hCTCF cassette at 0 kb (EC0) and 100 kb (EC100) (EC-lines). As expected, 4C-seq demonstrated that the ectopic CBSs served as boundaries: they hampered gene promoter and enhancer contacts across the integration sites but stimulated their contacts with downstream sequences to effectively create a smaller and more self-interacting domain containing the enhancer and reporter gene (Fig. 3c). Taking advantage of FRT sites flanking the μLCR, we removed the enhancer in EC0 and EC100 to create C0 and C100 cell lines (C-lines), which have only ectopic CBS (no enhancer) integrated in *cis* with the reporter gene. The C-lines showed that the ectopic CBS itself had no intrinsic transcription activation capacity and it did not stimulate the reporter gene to engage in new chromatin contacts as the enhancer did (Fig. 3d). The integrated CBS itself did form contacts with a convergent endogenous CTCF site (termed the 'left boundary site'), but this specific interaction and particularly its contacts with sequences elsewhere in the domain were strongly stimulated by the co-integrated enhancer, which was most notable in the EC0 line but was also appreciable in the EC100 line (Fig. 3d and Extended Data Fig. 4a). We then asked whether a 3×CTCF cassette at the gene, instead of the enhancer, had a similar impact. We created an E100 cell line (C-0E100) with a 3×CTCF cassette located downstream of and convergent to the gene. In C-0E100, the CBS gave some but much less support to transcription than in EC100, in which the CBS was placed upstream of the enhancer (Fig. 3e, compare with Fig. 3c (C-0E100 versus EC100)). We attributed this to the fact that the CTCF sites downstream of the gene failed to find a proximal looping partner, with the gene and enhancer consequently still acting in a larger and less insulated domain (Fig. 3e). However, we also considered the option that a CBS may act differently when flanking a gene or an enhancer. To further investigate this, we placed a dsRed reporter gene instead of the enhancer at the 100 kb position, in a dual reporter cell line that also had the original GFP reporter gene at 0 kb and the enhancer downstream at respectively 50 kb and 150 kb of each of the genes. We then deleted a nearly 400-kb genomic interval to linearly recruit the upstream endogenous strong CTCF boundary, placing it at position 100 kb, immediately flanking the dsRed reporter gene (Fig. 4a). Here, the juxtaposition of CBS to the gene strongly increased the enhancer-mediated transcriptional output per cell and the ability to maintain GFP expression over time (Fig. 4b,c). Thus, no matter whether it flanked the gene or the enhancer, a CBS introduced at position 100 that placed the E-P pair in a much smaller contact domain, strongly supported long-range enhancer-mediated transcription, while the CBS placed near the gene at position 0 that had little impact on the domain size, did not obviously support transcription. We therefore conclude that CTCF sites near enhancers or promoters can enable distal enhancers to confer increased transcriptional activity and stability to genes, when they substantially reduce the domain size. We speculate that this is because the CTCF sites concentrate intra-domain loop extrusion activity, but it is possible that they prevent the enhancer or promoter from engaging in competing distal contacts.

**Enhancer recruits cohesin to form domains and CTCF loops.** The formation of contact domains relies on cohesin, and the degree of self-interaction is believed to reflect the local loop extrusion activity[12,19]. Tissue-specific enhancers have been proposed to recruit cohesin[20,24,26,27,29–32]. To directly test whether our enhancer recruited cohesin to induce intra-domain chromatin interactions, we used CRISPR interference (CRISPRi) to deplete the cohesin components SMC1A and RAD21 in the E100, EC100 and E407 clones. In all clones, following depletion, we observed a loss specifically of the enhancer-induced chromatin contacts (Fig. 5a–d and Extended

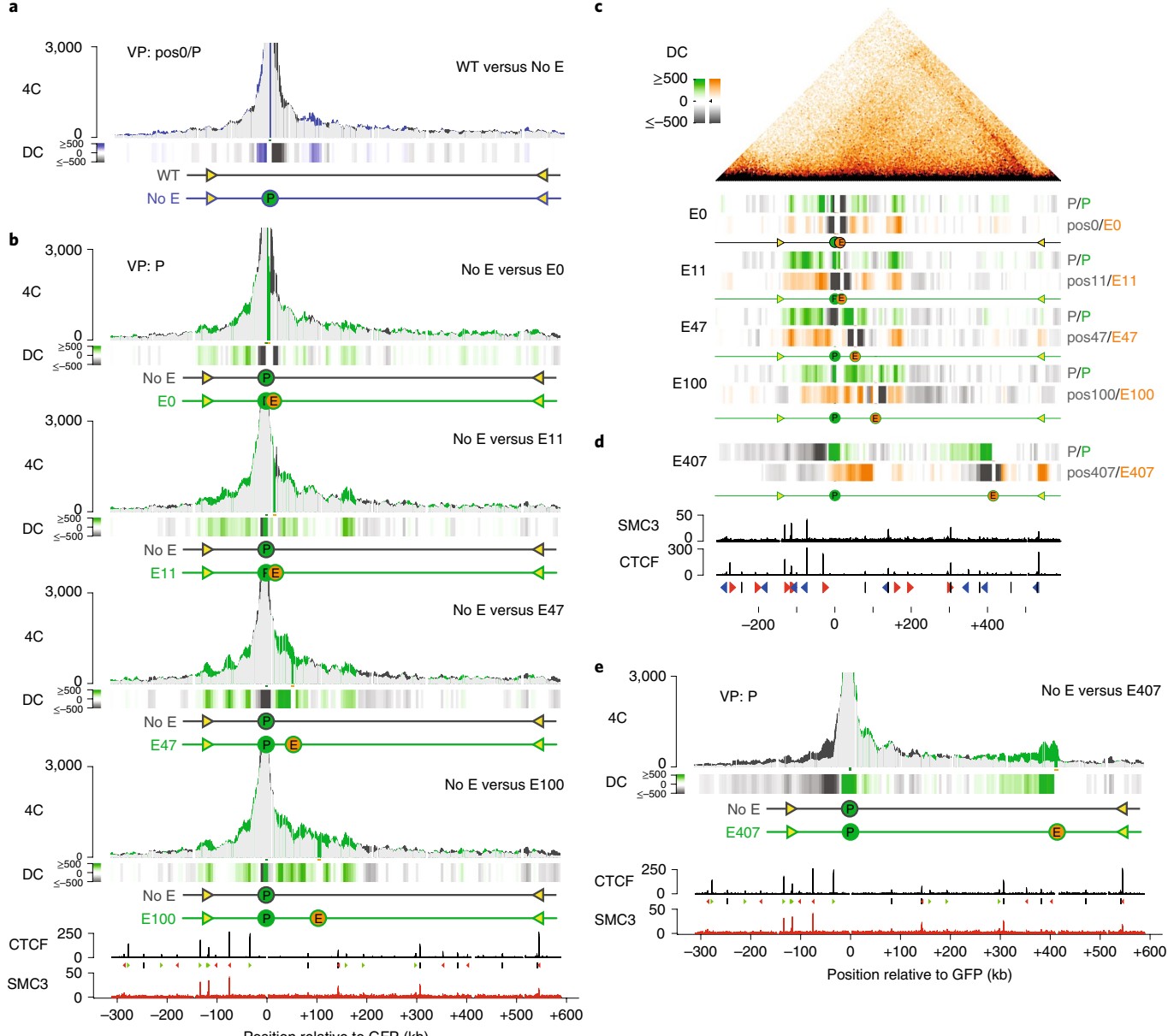

**Fig. 2 | Tissue-specific enhancer forms local self-interacting chromatin domains. a**, 4C-seq contact profiles plotted as overlays, comparing contacts of the integrated reporter gene promoter (P) in a cell line lacking the enhancer (no E, in blue), versus contacts of the corresponding endogenous genomic position (pos0) in wild-type (WT, in dark gray) K562 cells. Shared contacts are in light grey. y axis: 4C coverage per 1 million *cis*-reads. VP: 4C-seq viewpoint. The track below the graph shows the differential 4C-seq signal (DC: differential contacts), with contacts gained and lost by the gene promoter in blue and gray, respectively. $n = 2$ technical replicates/clone. **b**, 4C-seq profile overlays comparing contacts of the integrated promoter (P) in the different E-lines (green profiles), versus those of the integrated promoter in the cell line lacking an integrated enhancer (no E: dark gray profiles). Differential contacts are plotted below each overlay profile. Bottom, CTCF and SMC3 ChIP–seq tracks. **c**, 4C-seq differential contacts tracks showing, per E-line (E0, E11, E47 and E100), the gained (green) and lost (gray) contacts of the gene promoter (P) as compared with its contacts measured by 4C-seq in the cell line lacking the enhancer (top), as well as the gained (orange) and lost (gray) contacts of the enhancer (bottom), as compared with the contacts of its corresponding endogenous chromosomal position (pos0, pos11, pos47 and pos100, respectively) in the cell line lacking the enhancer. For reference, the aligned HiC contact map is shown on top. **d**, Same as in **c**, but for the E407 cell line. SMC3 and CTCF ChIP–seq signal and CTCF-site orientations are indicated at the bottom. **e**, 4C-seq overlay contact profiles showing gene promoter contacts gained (green) and lost (black) upon integration of the enhancer at position E407, as compared with its contacts in a cell line carrying no integrated enhancer.

Data Fig. 5), strongly supporting the hypothesis. To further validate whether the enhancer recruited cohesin to the locus, we performed quantitative-PCR-based chromatin immunoprecipitation (ChIP–qPCR) for SMC1A and analyzed cohesin levels at selected sites. In all E-lines with enhancers at varying distances from the reporter gene, except for the E407 line, we found more cohesin deposited at the

reporter gene promoter than in the control line lacking an enhancer (Fig. 6a). Without the μLCR, but with proximal (C0) or distal (C100) integration of the 3×-hCTCF cassette, little cohesin accumulation was observed at the promoter (Fig. 6a). We then asked whether the enhancer also accumulated cohesin on endogenous CTCF sites. In EC0 and EC100 cells, a strong, enhancer-stimulated chromatin loop

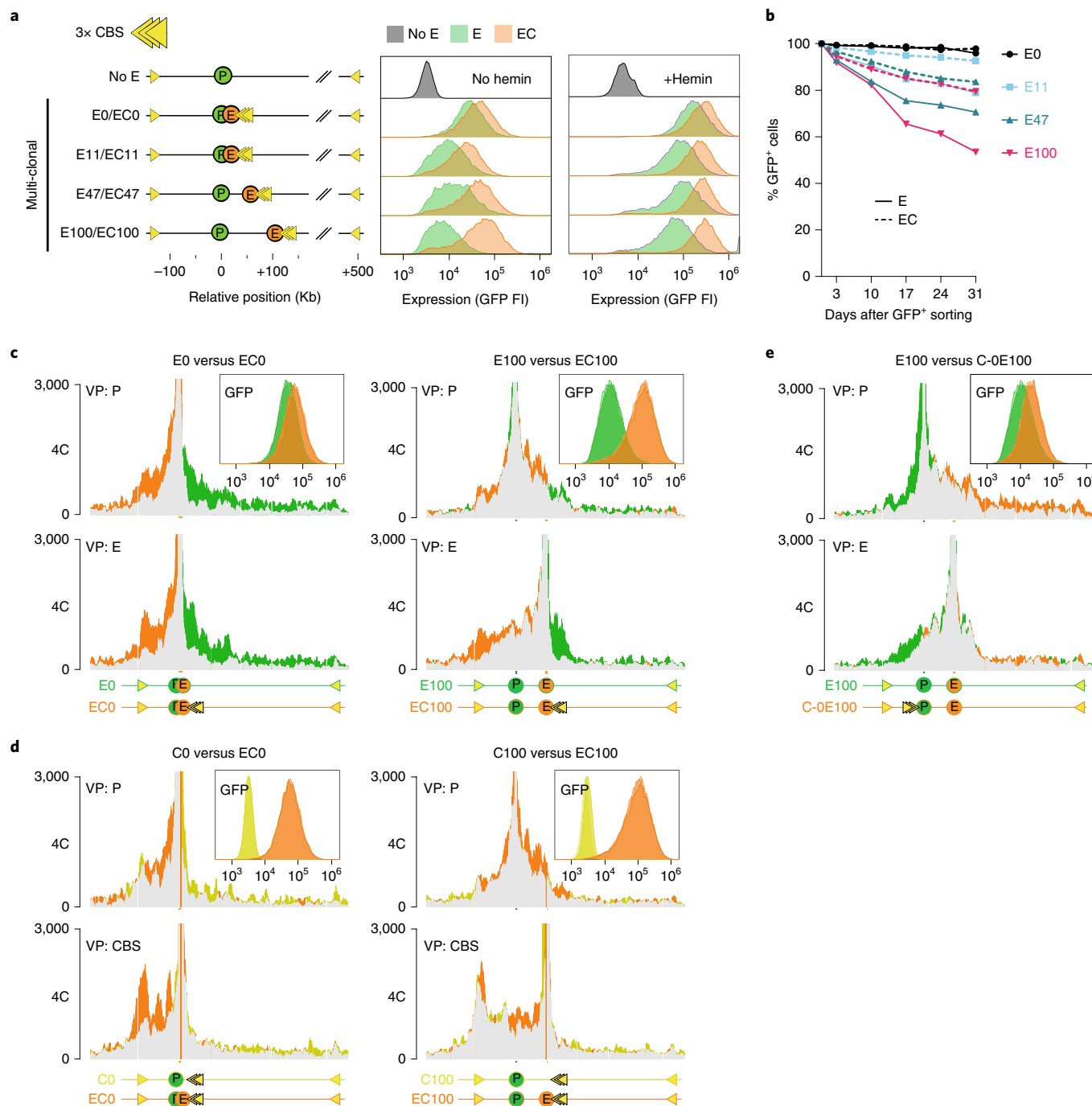

**Fig. 3 | Activated CTCF sites that reduce domain size support enhancer-dependent expression levels and stability. a**, GFP fluorescent intensity (FI) of multi-clonal (bulk) cell lines carrying the enhancer without (E, in green) and with (EC, in orange) the co-integrated 3×-hCTCF cassette, cultured without (left) or with hemin (right). Yellow triangles symbolize 3×-hCTCF. **b**, Percentage of remaining GFP-positive cells in multi-clonal (bulk) cell populations carrying the enhancer without (solid line) or with flanking 3×-hCTCF (dashed line) at the indicated distances. At each time point, cells were first treated with hemin for 2 days prior to FACS analysis. **c**, 4C-seq contact profile overlays, comparing contacts of the integrated gene promoter (P) and μLCR enhancer (E) in the E0 versus EC0 cell line (left), and in the E100 versus EC100 cell line (right), highlighting in orange their gained contacts and in green their lost contacts due to the presence of the 3×-hCTCF binding sites (CBS). y axis: 4C coverage normalized per 1 million *cis*-reads. *n* = 2 technical replicates/clone. VP: 4C-seq viewpoint. Also shown at the top right is the overlay of their FACS profiles, demonstrating that 3×-hCTCF at position 0 mildly supports enhancer-mediated GFP expression. *n* = 1 replicate per time point per clone. **d**, 4C-seq contact profile overlays comparing contacts of the integrated gene promoter (P, top) and 3×-hCTCF (CBS, bottom) in the EC0 versus C0 cell line (left) and the EC100 versus C100 cell line (right), with contacts gained and lost due to the enhancer shown in orange and yellow, respectively. *n* = 2 technical replicates/clone. Top right plots show the overlay of their FACS profiles, demonstrating that both in C0 and in C100, the 3×-hCTCF cassette itself does not activate GFP expression. **e**, 4C-seq contact profile overlays comparing contacts of the integrated gene promoter (P, top) and enhancer (E, bottom) in the C-0E100 versus E100 cell line, with contacts gained and lost due to the presence of 3×-hCTCF sites shown in orange and green, respectively. *n* = 2 technical replicates/clone. Top right plots show the overlay of their FACS profiles, demonstrating that the downstream 3×-hCTCF cassette only mildly supports enhancer-mediated GFP expression.

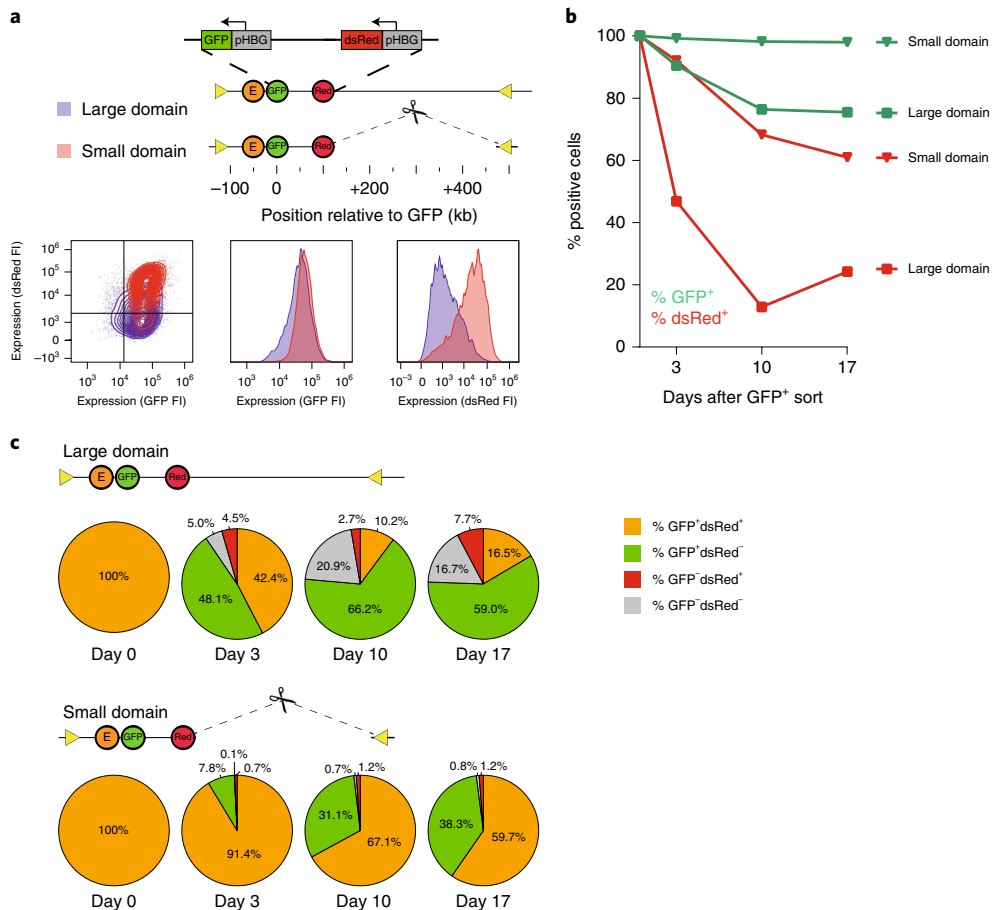

**Fig. 4 | An immediately flanking CTCF site helps a distal gene to compete for a shared enhancer. a**, Schematic representation of the two dual reporter lines. The μLCR enhancer was integrated 50 kb downstream of the GFP reporter gene, while a dsRed reporter gene driven by the same HBG1 promoter was integrated 100 Kb upstream of the GFP reporter gene. To place the genes and enhancer in a smaller domain, the 400-kb region between the dsRed reporter gene and the right domain boundary was deleted. Plots show hemin-induced expression (fluorescent intensity, FI) of both genes measured by FACS in the large domain cell line (purple) and the small domain cell line (red), 3 days after sorting each line for double-positive (GFP+dsRed+) cells. Lower left panel highlights that, in the context of a small domain, more cells manage to keep both genes active, and expressed at higher levels. Middle panel shows GFP expression (FI) for both clones, and right panel shows dsRed expression (FI) for both clones, highlighting that particularly the expression of the distal dsRed reporter gene benefits from being part of a smaller domain. **b**, The small domain protects against gene silencing, particularly of the distal dsRed reporter gene. Percentage of remaining GFP-positive (green lines) and dsRed-positive cells (red lines) after sorting the 'small domain' cells (triangles) and the 'large domain' cells (squares) each for double positive (GFP+dsRed+) cells and culturing them for the indicated period. At each time point, cells were first treated with hemin for 2 days prior to FACS analysis. n = 1 replicate per time point per clone. **c**, Transgene silencing is not the consequence of enhancer inactivation. Percentages of double-positive (orange), GFP-positive (green), dsRed-positive (red) and double-negative (gray) cells at 0, 3, 10 or 17 days after double-positive sorting of the indicated clones, as measured by flow cytometry.

was observed between the integrated 3×hCTCF cassette and a convergent endogenous CTCF site that we termed the 'left boundary site' (Fig. 3d). We focused on this endogenous CTCF site. Because it is also present on three untargeted copies of chromosome 18 (K562 has four copies of our genomic segment[47]), exclusive analysis of its characteristics on the targeted allele was not possible. Despite this technical limitation, 4C-seq directed to this position confirmed that this normally unengaged CTCF site now formed a novel chromatin loop with the integrated CTCF sites in both cell lines (Extended Data Fig. 4a,b), and ChIP–qPCR demonstrated that the integrated enhancer deposited cohesin at this left boundary site (Fig. 6b). Finally, we investigated cohesin levels at our integrated CTCF sites in cell lines with and without a co-inserted enhancer. The ectopic CTCF sites already recruited cohesin in the C-lines, but also cohesin accumulated to higher levels in both EC-lines, having the enhancer co-integrated with the CBS (Fig. 6c). In contrast, binding of the CTCF protein to these ectopic CTCF sites, and to the endogenous

left boundary site, was not controlled by the enhancer (Fig. 6d,e). From the chromatin topology studies in cohesin-depleted cells and the ChIP–qPCR results, we concluded that the enhancer recruited extruding cohesin complexes to the locus and deposited them at flanking convergently oriented CTCF molecules.

To further investigate the domain-forming capacity of the enhancer, we created two cell lines, again through a series of consecutive genetic modifications, with an identically modified CTCF binding landscape, but with a differently located enhancer. For this, the 3×-hCTCF cassette was placed at position 100 (C100), as before, and another CTCF cassette (3×-mCTCF, with three strong CTCF binding sites selected from the mouse genome[48], that is sequences not existing in the human genome) was placed in identical orientation at position 0 (C0). One line then carried the μLCR downstream of the one CTCF cassette (EC0–C100), while the other line had the enhancer downstream of the other CTCF cassette (C0–EC100). When we applied 4C-seq to the endogenous left boundary

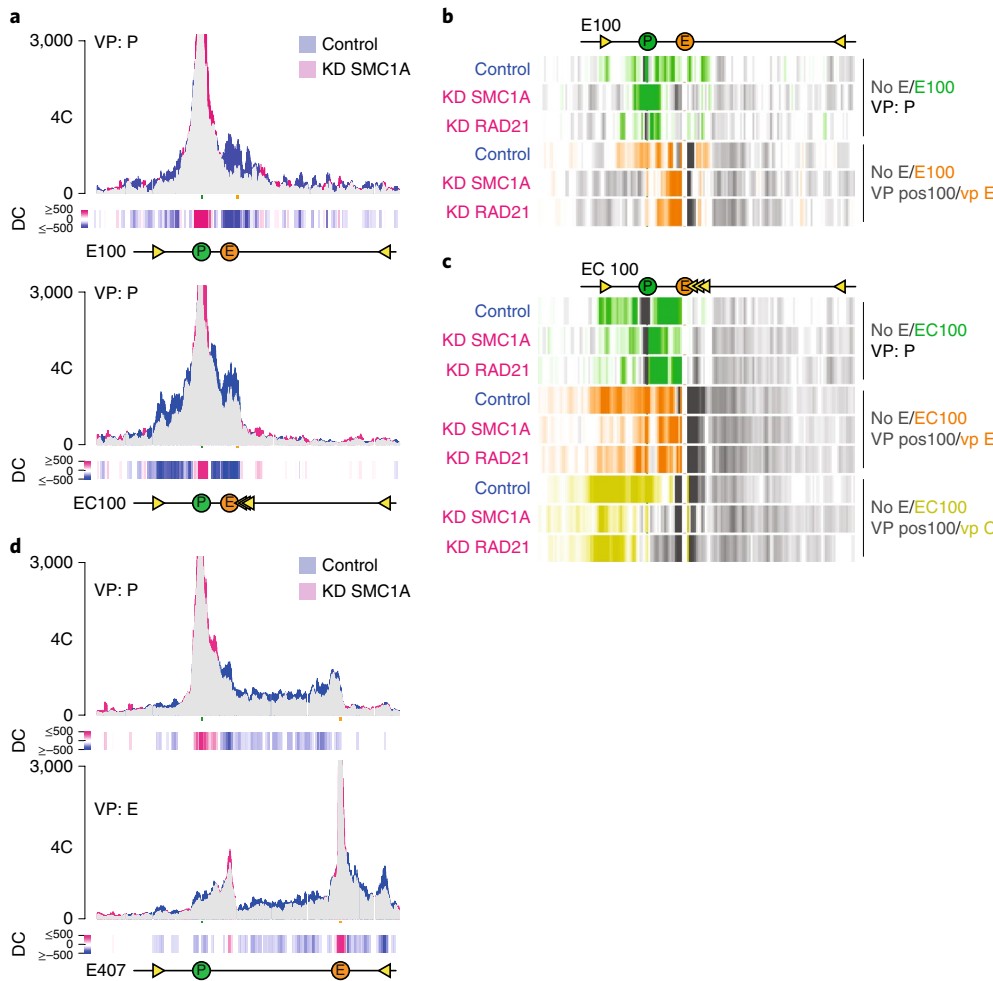

**Fig. 5 | Tissue-specific enhancer requires cohesin to create self-interacting domains and activate CTCF sites. a**, SMC1A depletion causes loss of enhancer-induced contacts. 4C-seq contact profile overlays comparing contacts of the integrated promoter (P) in two cell lines, E100 (top) and EC100 (bottom), following CRISPRi knockdown (KD) of SMC1A (pink) versus contacts measured in cells treated with control single guide RNA (sgRNA) (blue). Shared contacts are in gray. *y* axis: 4C coverage normalized per 1 million *cis*-reads. Differential contacts plotted per 5-kb genomic bins are shown below. **b,c**, 4C-seq differential-contacts tracks showing, for cell lines E100 (**b**) and EC100 (**c**), in green the gained contacts of the gene promoter in control cells, SMC1A-depleted cells and RAD21-depleted cells, as compared with its contacts in the cell line lacking the enhancer. Similarly, in orange and yellow, respectively, are shown the gained contacts by the enhancer and the 3×-hCTCF sites in control cells, SMC1A-depleted cells and RAD21-depleted cells, as compared with the contacts of their corresponding endogenous chromosomal position in the cell line lacking the enhancer (no E). Note that in all instances, the enhancer-stimulated contacts seen in control cells are (partially) lost again upon cohesin depletion. **d**, SMC1A depletion causes loss of enhancer-induced contacts in E407. 4C-seq contact profile overlays comparing contacts of the integrated promoter (P, top) and enhancer (E, bottom) in E407 cells following CRISPRi depletion of SMC1A (pink), versus those measured in E407 cells treated with control sgRNA (blue). *y* axis, 4C coverage normalized per 1 million *cis*-reads. Differential contacts (DC) plotted per 5-kb genomic bins are shown below.

site, we found that the E0 stimulated left boundary contacts mostly with C0, creating a small domain, whereas the E100 enabled the left boundary to select the distal C100 as its preferred contacting partner (Extended Data Fig. 4b). Also, the reporter gene made very different contacts in the two cell lines with identical CTCF binding landscapes, with the distally integrated enhancer (E100) activating the gene to contact a much larger domain (Fig. 6f). Thus, the location of the enhancer determined the domain that was formed. ChIP–qPCR confirmed again that the enhancer recruited cohesin for deposition at flanking convergent CTCF sites and at the reporter gene promoter (Fig. 6g). Our data therefore demonstrated that the tissue-specific enhancer created local self-interacting domains and CTCF-mediated chromatin loops through the recruitment of extruding cohesin complexes. The location of the enhancer dictated which flanking convergent CTCF sites were selected for cohesin stalling and for the formation of domain-spanning chromatin loops.

**Cohesin needed for distal but not proximal enhancer action.** Knowing that the enhancer recruits cohesin to mobilize the gene, we next studied cohesin's requirement for enhancer-mediated gene activation. We took all E-lines with the enhancer at varying distances from the target gene. In the distal configurations (E100 and E407), knockdown of all three cohesin subunits, RAD21, SMC3 and SMC1A, led to strong down-regulation of reporter-gene expression (Fig. 7a–c). At E47, knockdown of these factors had a less negative impact on expression, while at the two most proximal E-P combinations, E0 and E11, knockdown of cohesin subunits had the opposite, namely positive, effect on gene expression. To relate this to the impact of other factors, we also knocked down GATA1, a transcription factor known to support transcription and looping in the β-globin locus[49], and two components (MED1 and MED21) of mediator[50], a protein complex important for enhancer function. Different from cohesin, GATA1 and mediator were required for transcriptional

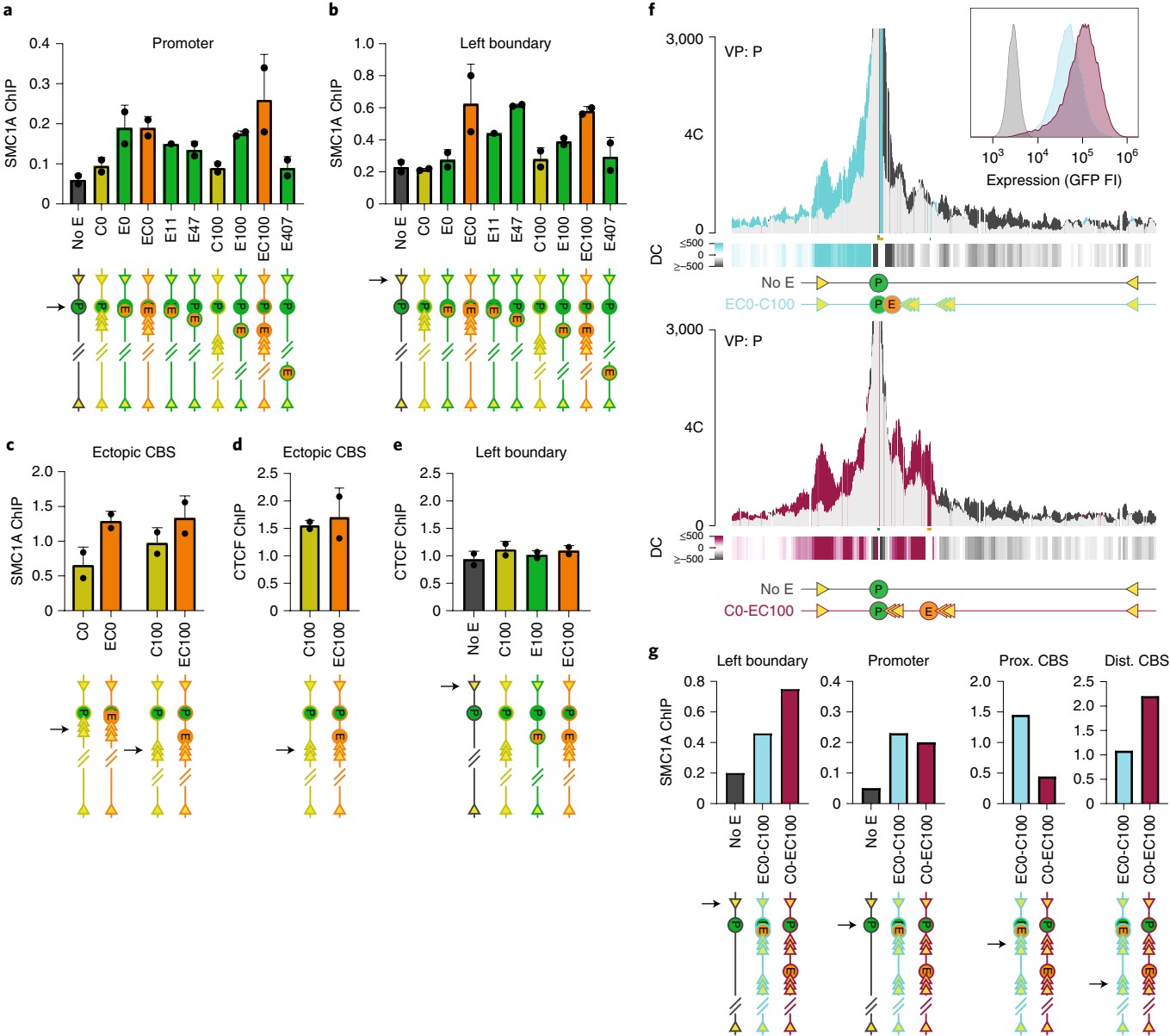

**Fig. 6 | Tissue-specific enhancer recruits cohesin to create self-interacting domains and activate CTCF sites. a–e**, The enhancer deposits cohesin (**a–c**) but not CTCF (**d,e**) at flanking CTCF sites and the gene promoter. ChIP–qPCR results showing relative enrichment of SMC1A protein bound to the integrated GFP reporter gene promoter (**a**), the left boundary site (**b**) and 3×-hCTCF binding sites (**c**), and of CTCF bound to 3×-hCTCF binding sites (**d**) and left boundary (**e**). Enrichment values on the *y* axis are relative to that measured at a strong SMC1A/CTCF binding site in K562 cells (see Methods). Data are presented as mean values ± s.d. *n* = 2 biological replicates per clone per position. **f**, The location of the enhancer dictates the domain contacted by the gene. Top: 4C-seq contact profile overlay showing the contacts of the integrated reporter gene promoter (P) gained (in blue) when located in the indicated CTCF binding landscape with an immediate proximal enhancer (E0). Bottom: 4C-seq contact profile overlay showing the contacts of the integrated reporter gene promoter (P) gained (in red) when located in the same CTCF binding landscape, but with the enhancer at position E100. In both plots, contacts are compared with that of the promoter in a cell line (no E) lacking integrated enhancers or CTCF sites. *y* axis: 4C coverage normalized per 1 million *cis*-reads. Differential contacts plotted per 5-kb genomic bins are shown below. *n* = 2 technical replicates/clone. *n* = 1 replicate per clone per position. **g**, The enhancer determines which CTCF sites are selected for cohesin deposition. ChIP–qPCR enrichment for SMC1A at the left boundary, promoter, proximally integrated 3×-mCTCF binding sites and distally integrated 3×-hCTCF binding sites, for a cell line without enhancer (no E, dark gray), a cell line with an enhancer and CTCF sites directly upstream of the promoter and additional CTCF sites 100kb upstream (EC0–C100, blue) and a cell line with CTCF sites directly upstream of the promoter and the enhancer and CTCF sites integrated 100kb upstream (C0–EC100, red).

activation irrespective of enhancer distance to the promoter (Fig. 7a,b). Thus, the enhancer strictly needed transcription factors and mediator for its activity, but relied on cohesin only for the activation of distal genes, not for the activation of proximal genes. This strongly suggested that an enhancer requires cohesin loop extrusion to bring a distal gene in proximity in order to activate it.

## Discussion

By using a unique, bottom-up approach of building many different regulatory landscapes in an inactive chromatin environment, we provide experimental evidence for an emerging model[20,24,26,27,31,51,52] in which the concerted action of tissue-specific transcription factors at enhancers serves two purposes: (1) they enable recruitment

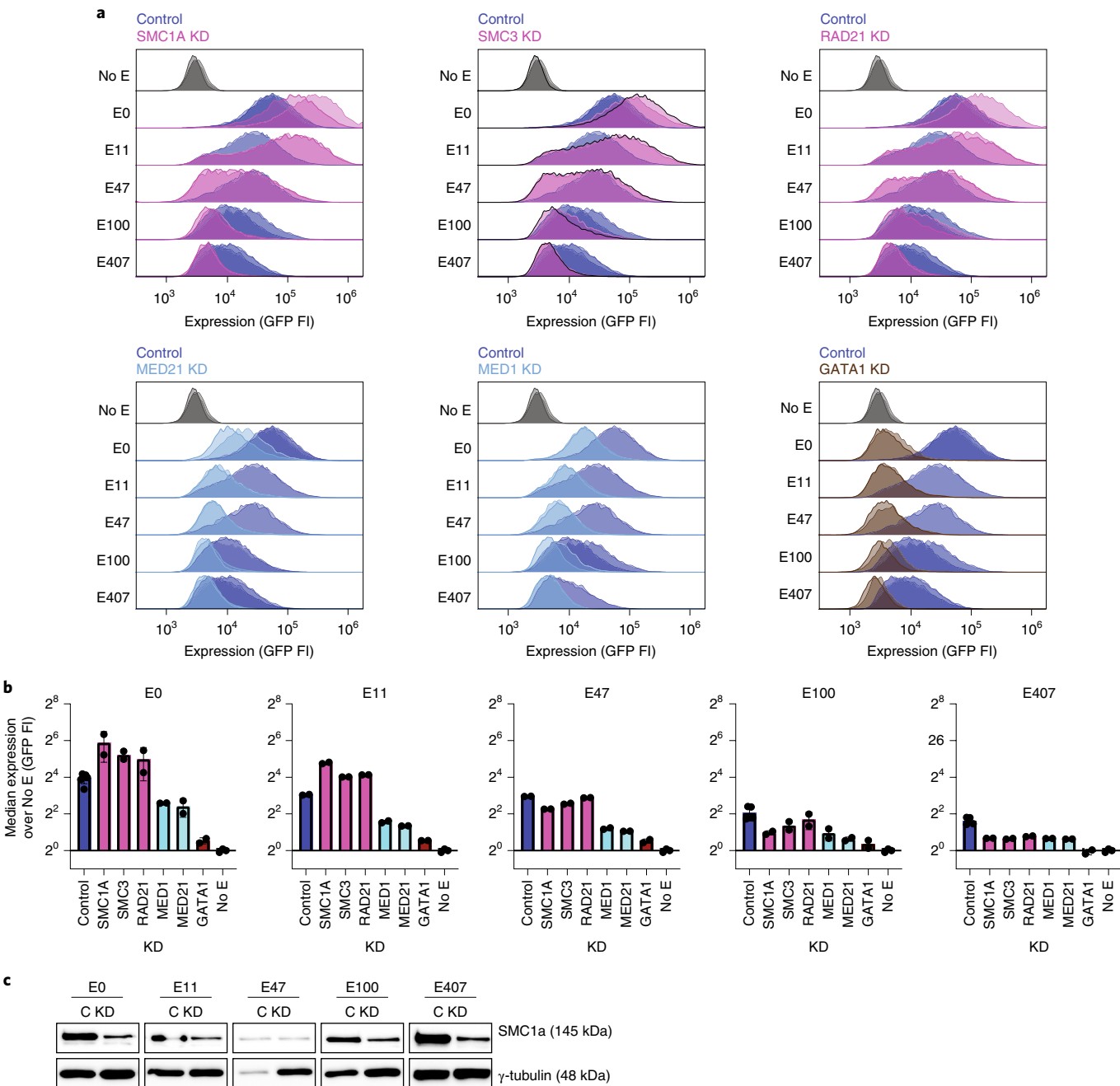

**Fig. 7 | Cohesin is required for long-range gene activation but dispensable for short-range enhancer action. a**, GFP expression measured by FACS (FI) in arbitrary units in E0, E11, E47, E100 and E407 cells depleted by CRISPRi of the cohesin subunits SMC1A, SMC3 and RAD21 (top, in pink), the mediator components MED21 and MED1 (bottom left, in light blue) and the transcription factor GATA1 (bottom right, in brown) as compared with GFP expression in cells treated with a control sgRNA (all panels, dark blue). For comparison, the top rows show GFP expression in the cell line lacking an integrated enhancer (no E cells: dark gray). Data in all histograms come from at least two replicates. **b**, Median GFP expression measured by FACS (FI) in the different cell lines upon CRIPSRi-mediated knockdown of cohesin components SMC1A, RAD21 and SMC3 (pink), mediator components MED21 and MED1 (light blue) and transcription factor GATA1 (brown) or control (dark blue), plotted relative to the median expression measured in cells without an enhancer. Data are presented as mean values ± s.d. of at least two replicates. Note that independent of its distance to the reporter gene promoter, enhancer activity is severely affected upon loss of MED21, MED1 and GATA1, while cohesin is exclusively required for long-range enhancer action, not for activation of an immediately proximal gene promoter (E0). **c**, Western blot confirming knockdown (KD) of cohesin subunit SMC1A in E0, E11, E47 and E407. Control (C) is the same cell line treated with a control sgRNA. SMC1A KD efficiency was assessed once per clone and condition.

of co-factors, together with which they can stimulate initiation and elongation of the transcriptional machinery at gene promoters, and (2) they enable recruitment of cohesin, which, presumably through DNA extrusion, locally stimulates looping and contacts with and between more distal sequences, to form contact domains. We here show that cohesin recruitment is necessary for enhancers to activate distant, but not proximal, genes. Since cohesin forms chromatin loops, this supports the debated idea that long-range gene activation, certainly in an inactive chromatin context, requires chromatin looping. Importantly, developmental genes often rely on distant

tissue-specific enhancers located in inactive chromosomal regions. Interestingly, when separated over short linear distances (E0, E11, and even E47) our E–P pair functioned perfectly well (or even better) under conditions of reduced cohesin. This and similar observations made in a parallel study on the *Shh* locus[53] argue that we may need to better define the meaning of 'long range' when discussing mechanisms of gene activation. We postulate that the type (strength) of the enhancer and the genomic context determines whether the enhancer relies on cohesin-dependent or cohesin-independent mechanisms[25,28] to fully activate a target gene over intermediate distances (approximately 0–100 kb). Molecular condensates formed through non-specific interactions between intrinsically disordered domains of enhancer- and promoter-associated transcription factors and mediator may enable cohesin-independent E–P communication over such distances[54–57]. Over farther distances, we expect that enhancers increasingly need cohesin for gene activation.

We further find that the genomic location of the enhancer determines which flanking pair of convergent CTCF sites is selected as boundaries of the contact domain. If boundaries immediately flank the gene or enhancer and place them in a small domain, they can support the distal enhancer in conferring strong and stable expression to the target gene. We propose this is a consequence of concentrated loop extrusion activity. We, like others, find that transcriptional activity is controlled by more than just E–P contact frequencies[36,45,46]. Preformed contacts established prior to gene activation have been observed at, for example, the *Hoxd* locus, and such 'permissive' structures[58] may be reminiscent of the topologies that we detect here. It was striking to see that gene silencing or hyper-activation (by stimulating cells with hemin) only mildly decreased or increased (respectively) E–P contact frequencies, as measured by 4C-seq. Importantly, our data consistently showed that the enhancer stimulated the distal gene not only in E–P contacts, but also in contacts with intervening and immediately surrounding sequencing. This may suggest that recruited cohesin does not stay anchored at the enhancer, but extrudes and simultaneously migrates away from its entry site. Interestingly, modeling showed that local chromatin loops can stimulate E–P communication, even if they are not directly anchored at the regulatory elements[59].

Finally, a particularly clear E–P distance effect was seen when we followed expression over time in our cell cultures: the further apart on the chromosome, the more difficult it was for the enhancer to protect the target gene from silencing. Silencing most likely was polycomb-mediated and (initially) took place at the reporter gene promoter, not at the enhancer. We concluded the latter from cell lines with two reporter genes controlled by a shared enhancer. Almost always, it was either one (most often the distal) or the other reporter gene that silenced, but not both (Fig. 3h), as would be expected if the enhancer was silenced. The enhancer remaining active may also explain why non-expressing cells still showed enhancer-induced topological features and would be in line with its definition as a 'locus control region'[60].

In summary, our work demonstrates that an enhancer can recruit cohesin to stimulate the formation of a self-interacting domain, engage flanking CTCF sites in loop engagement and activate expression of distal target genes.

## Online content

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

## Methods

**Statistics and reproducibility.** Technical replicates were done for all experiments. All main conclusions are based on observations reproducibly made across multiple independent clones, that is clones with the enhancer, reporter gene and/or ectopic CTCF sites integrated at different locations. No statistical method was used to predetermine sample size, no data were excluded from the analyses, the experiments were not randomized and the investigators were not blinded to allocation during experiments or outcome assessment.

**Cell culture.** Human erythroleukemia K562 cells were used in this study (not an authenticated cell line; available at our institute; periodically tested for mycoplasma). Cells were grown at 37 °C at 5% $CO_2$ in RPMI 1640 (Gibco) with 10% FBS (Sigma) and 1% penicillin–streptomycin (Gibco). Density was kept between $1 \times 10^5$ and $5 \times 10^5$ cells per ml medium. Cells were routinely tested for mycoplasma. Prior to every experiment, cells were weekly bulk-sorted for 2 consecutive weeks on Becton Dickinson SORP FACSAria FUSION Flow Cytometer. Gating was based on cells without an enhancer (no-E cells) or fluorescence-minus-one controls. Flow cytometry analysis was done 3 days after sorting on a Beckman Coulter Cytoflex S. For differentiation, culture medium was supplemented with porcine hemin (Sigma, 30 μM final concentration) 1 day after sorting. Hemin and medium were refreshed 1 day later. We prepared 4 mM hemin stock solutions according to ref. [62] and kept them at −20 °C. For long-term culturing and FI monitoring, at each indicated time point an aliquot of cells was taken and treated with hemin for 2 days, and FI was monitored. To obtain irreversibly silenced populations of cells, GFP-negative cells were weekly sorted for 6 (E100) or 9 (E407) consecutive weeks.

**Targeting constructs.** Targeting constructs contained one or two regulatory elements (μLCR[41], μLCR–3×mCTCF[48], μLCR–3×hCTCF (described below), dsRed–pHBG[63]) of interest flanked by two ~1-Kb-sized site-specific homology arms that were amplified from genomic DNA with primers as indicated in Supplementary Table 1. Fragments were combined into a plasmid by In-Fusion Cloning (Takara).

The three human CTCF sites (3×hCTCF) were selected from ref. [61] and PCR-amplified from the following genomic DNA positions: chr19:41650330-41650595, chr7:39559582-39559824 and chr13:21498993-21499294. These were combined by overlap-extension PCR using primers as indicated in Supplementary Table 1.

**Clonal cell line generation.** The founder cell line containing the d2eGFP reporter driven by human pHBG1 and μLCR enhancer was generated by Tol2 transposition. The founder μLCR is flanked by sleeping beauty (SB) terminal inverted repeats (ITRs) and LoxP sites which split a puromycin gene driven by pSV40 such that it is not functional. GFP-expressing cells upon transgene transposition were single-cell sorted, and integration sites were mapped using TLA[64]. The μLCR enhancer was removed by transient transfection of a CRE-recombinase encoding plasmid. The clone that had the transgene integrated at chromosome 18 (position: 19609009) was selected for further experiments and designated 'no E'.

Regulatory elements were integrated in target cell lines using CRISPR–Cas9 and homology-directed repair. Cas9 plasmid (pSpCas9(BB)-2A-BFP (a modified version of PX458 (Addgene plasmid no. 48138), in which eGFP is replaced for tagBFP)) containing a position-specific single guide was co-transfected with a targeting plasmid, and GFP-expressing cells were single-cell sorted using flow cytometry, expanded, genotyped and Sanger-sequenced for correct integration of the transgene. Guide sequences specific for position of interest were cloned between BbsI sites and are listed in Supplementary Table 2.

The μLCR enhancer was targeted to no-E cell line to positions 11 kb and 47 kb upstream of the eGFP reporter to obtain the respective clonal cell lines. μLCR inserted at position 407 kb carried an additional Anch4 imaging platform (kind gift from K. Bystricky). μLCR–3×hCTCF was targeted to positions 0 Kb (pos0) and 100 Kb (pos100) to obtain EC0 and EC100, with loxP sites flanking the 3×-hCTCF cassette and FRT sites flanking the μLCR. Additionally, μLCR–3×mCTCF was targeted to no E to obtain EmC0. E0 and E100 were generated by transient transfection of a CRE-recombinase (Cre) encoding plasmid. C0, mC0 and C100 were made by expressing the recombinase Flippase (Flp) in the respective EC-lines. The C0-EC100 clonal cell line was generated by targeting μLCR–3×hCTCF to pos100 in cell line mC0. For EC0-C100, μLCR×3×mCTCF was targeted to C100, pos0. C-0 E100 was generated by deleting the 3′ part of eGFP and subsequently repairing it with a targeting construct containing 3×hCTCF–eGFP flanked by homology arms. Homology arm primers and genotyping primers are indicated in Supplementary Table 1.

To place the E407 enhancer immediately upstream of the GFP reporter gene, the genomic region between the integrated eGFP and the integrated μLCR at position 407 kb was removed by transient transfection of two Cas9 plasmids (pSpCas9(BB)-2A-BFP), each containing a specific single guide targeting the upstream or downstream sites of this region. Guides are listed in Supplementary Table 2.

Double reporter cell lines were generated by targeting dsRed–pHBG1 to pos100 in a separately generated E50 cell line that also had a flanking Anch4 sequence. GFP$^+$dsRed$^+$ double-positive cells were selected to obtain single clones.

The intervening region between dsRed reporter and right boundary was removed as described above.

**Multi-clonal cell line generation.** μLCR or μLCR–3×hCTCF was targeted as for the clonal cell lines on the same day to noE cell line at positions 0 Kb, 11 Kb, 47 Kb and 100 kb upstream of the eGFP reporter to obtain the respective E- and EC-lines. After 5 days, GFP-expressing cells were bulk-sorted weekly for 2 consecutive weeks and analyzed in the presence or absence of hemin 3 days after the second sort. Cells that were not treated with hemin were kept in culture and induced with hemin prior to analysis at days 10, 17, 24 and 31 after the second sort.

**KRAB silencing.** For gene knock-down experiments, dead-Cas9 (dCas9) fused to Krüppel-associated box (KRAB) was randomly integrated in the genome of the target cell line via lentiviral transduction of a modified version of pHR-SFFV-dCas9-BFP-KRAB (Addgene plasmid no. 46911) carrying a P2A blasticidin selection and UCOE element (kind gift of M. Tanenbaum). The top 50% BFP-expressing cells were bulk-sorted and kept under 0.7 μg/ml blasticidin selection (Sigma). sgRNA sequences against genes of interest were cloned into lentiviral targeting plasmid pU6-sgRNA EF1Alpha-puro-T2A-BFP (Addgene plasmid no. 60955). Optimal target sequences were selected from ref. [65]. dCas9–KRAB-containing target cells were first sorted for GFP for 2 consecutive weeks and transduced with sgRNA-coding virus 2 days after the second sort. Cells recovered for 1 day before puromycin selection (Sigma, 1 μg/ml final concentration) was started. Crosslinking for 4C and analysis by flow cytometry was performed 5 days following puromycin selection. sgRNA sequences are listed in Supplementary Table 2.

**Flow cytometry analysis.** Flow cytometry standard (FCS) files were acquired with CytExpert 2.3.0.84 and analyzed with FlowJo 10.5.2 software. Fluorescence compensation was applied on the basis of single-fluorophore samples. Cells were transferred to a 96-well plate for flow cytometry analysis directly from the culture dish. Cells were gated for live single cells on the basis of FSC-A, SSC-A and FSC-W. For every experiment, at least 10,000 cells within the live single-cell gate were recorded. For silencing experiments, cells were considered GFP-positive on the basis of gating on the no-E cell line treated with hemin. For KD experiments, cells were considered sgRNA-positive on the basis of control cells containing dCas9–KRAB but no sgRNA as indicated in Supplementary Information Figure 1. To calculate relative fluorescence, the median GFP FI of KD samples was divided by that of control samples after subtraction of GFP FI for the noE sample.

**Fluorescence-activated cell sorting.** For fluorescence-assisted cell sorting, ten million cells were collected, centrifuged and resuspended in 1 ml of culture medium. Cells were considered GFP positive on the basis of gating on the noE cell line. Gates were drawn such that more than 99.9% of noE cells fell into the GFP-negative gate. To enrich for KRAB–BFP-containing cells prior to knock-down experiments, cells were gated to be BFP-positive. From this population, the top 50% BFP-expressing cells were sorted. For analysis of knock-down experiments, cells were gated to be BFP sgRNA positive as compared with cells that contained only KRAB–BFP.

**4C-seq.** 4C-seq was performed as described[44]. Eight to ten million cells were cross-linked with formaldehyde. DNA was digested in situ with Csp6I (first cutter) and NlaIII (second cutter). Indexed Illumina sequencing adapters were introduced to ligation fragments of interest with a two-step PCR strategy. All viewpoint-specific primers are listed in Supplementary Table 1. Technical replicates were processed on the same day.

**4C analysis.** FastQ files were demultiplexed and processed as in ref. [44] (https://github.com/deLaatLab/pipe4C) Reads were mapped against versions of hg19 that were modified to contain the aforementioned insert sequences (modified genomes) at the experimentally validated coordinates.

Plotting and contacts counting was done in R (version 4.1.0, https://www.R-project.org/). Blind fragments were omitted for analysis. Read counts were then normalized to a million mapped intra chromosomal reads (normalized reads) excluding the two highest covered fragments and 21 fragment end rolling mean scores were calculated for every fragment end. For plotting of overlay profiles of distinct modified genomes, positions of fragment ends were shifted on the basis of the coordinates of the inserted sequences, such that common fragment ends were aligned.

For E-P contact counting, 4C profiles with viewpoint eGFP reporter were used. Reads normalized to 1 million *cis*-reads were counted for the 11 non-blind enhancer fragment ends that could be uniquely mapped using all modified genomes and averaged. Mean coverage per fragment was used for plotting.

For differential contact tracks, normalized reads per fragment were averaged per 5-kb bins for each profile and subtracted, which resulted in a number for the average differential contacts per fragment in the 5-kb bin. Differential contacts were then represented in a color range, where less than 50 normalized read differences were indicated in white and then color-scaled between 50 and 500 differential contacts, as indicated.

**ChIP–qPCR.** For each batch of ChIP experiments 2.5 million cells were sorted for GFP expression. Cells were cultured for 4 days to obtain at least 25 million cells, followed by fixation for 10 minutes at 4 °C in 1% PFA. From this point onward, cells were processed via the ChIP-IT High Sensitivity kit (Active motif) as per the manufacturer's instructions. Chromatin was sheared to 200- to 500-bp fragments on a Bioruptor Plus (Diagenode; 5 × 5 cycles of 30 s on and 30 s off, at the highest power setting). Immuno-precipitation was carried out by adding 5 μg of the appropriate antiserum (SMC1: A300–055A, bethyl; CTCF: 07-829, Millipore; H3K27me3: ab6002, Abcam) to approximately 30 μg of chromatin, which was incubating on a rotator overnight at 4 °C in the presence of protease inhibitors. Following addition of protein G agarose beads (SMC1) and magnetic G beads (CTCF, H3K27me3) and washing, DNA was eluted using DNA purification elution buffer (Active motif). The eluted DNA was used in qPCR (Bio-Rad CFX Manager 2.1) using primers targeting putative CTCF/cohesin bound positions and primers targeting the GFP reporter gene for cohesin and H3K27me3, respectively. Putative CTCF/SMC1a and H3K27me3 sites were normalized over a tested bound site (CTCF/cohesin: chr11:4658282-4658362; H3K27me3: chr1820763505-20763408). All primers are listed in Supplementary Table 1.

**Immunoblotting.** Protein lysates were obtained after incubation for 30 minutes with RIPA buffer (50 mM Tris-HCl Ph 7,5, 1 mM EDTA, 150 mM NaCl, 0.25% deoxycholic acid, 1% Nonidet P-40), 1 mM NaF and NaV$_3$O$_4$ and protease inhibitor cocktail (11873580001, Sigma Aldricht). Next, samples were centrifuged at full speed (12$g$) for 10 minutes, and supernatants were collected and processed by standard SDS–PAGE Immunoblot. The following immunoblot antibodies were used: SMC1 (Bethyl, A300-055A, 1:1000), γ-tubulin(GTU-88, Sigma, T6557, 1:3000), goat anti-rabbit HRP (Cell Signaling no. 7074s, 1:3,000) and goat anti-mouse HRP (Cell Signaling no. 7076s, 1:3,000). Membranes were developed using SuperSignal West Dura (Thermo Fisher). Source Data Fig. 1 shows unprocessed gels.

**Reporting summary.** Further information on research design is available in the Nature Research Reporting summary linked to this article.

## Data availability

Raw sequencing data and mapped wig files are available without restriction from the Gene Expression Omnibus (GEO) under accession GSE180566. Public ChIP–seq and RNA-seq (mapped to human genome release 19) tracks were downloaded from ENCODE portal[66]. The following data sets from ENCODE were used: K562 SMC3 ChIP-seq (Encode, M. Snyder, ENCSR000EGW, ENCFF479BWQ), ChIP–seq CTCF (Encode, B. Bernstein, ENCSR000AKO, ENCFF000BWF), K562 RNA-seq (Encode, B. Graveley, ENCSR000AEN, ENCFF657EOD, ENCFF578WIM). H3K27ac (Encode, B. Bernstein, GEO: GSM733656) and H3K27me3 (Encode, M. Snyder, GEO: GSM788088). Other data are available upon reasonable request Source data are provided with this paper.

## Code availability

Code for processing 4C data used in this study can be found at https://github.com/deLaatLab/pipe4C.

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

## Acknowledgements

We thank A. Allahyar for help with 4C analysis and de Laat lab members for discussions and feedback. This work is part of the Oncode Institute, and was funded by a VICI grant (724.012.003) and an NWO Groot grant (2019.012) from the Netherlands Organisation for Scientific Research (NWO), a Fondation Leducq (14CVD01) Transatlantic Network grant and an EU MSCA-ITN grant (ENHPATHY 860002).

## Author contributions

N. J. R., K. S., P. H. L. K. and W. d. L. conceived experiments; N. J. R., M. J. A. M. V., Y. O., C. V.-Q., A.-K. F., T. F., Z. d. A. d. R, P. H. L. K. generated constructs and cell lines; N. J. R. and S. v. d. E. performed flow cytometry; N. J. R. and S. J. D. T. performed KD experiments; N. J. R., S. J. D. T., M. J. A. M. V. and T. F. performed 4C experiments; N. J. R., R. H. and K. S. designed and performed ChIP-seq experiments; N. J. R. and P. H. L. K. performed bioinformatics analyses; N. J. R., K. S., P. H. L. K. and W. d. L. wrote the manuscript, with input from other authors.

## Competing interests

W. d. L. is a founder and shareholder of Cergentis B.V. The remaining authors declare no competing interests.

## Additional information

**Extended data** is available for this paper at https://doi.org/10.1038/s41594-022-00787-7.

**Correspondence and requests for materials** should be addressed to Wouter de Laat.

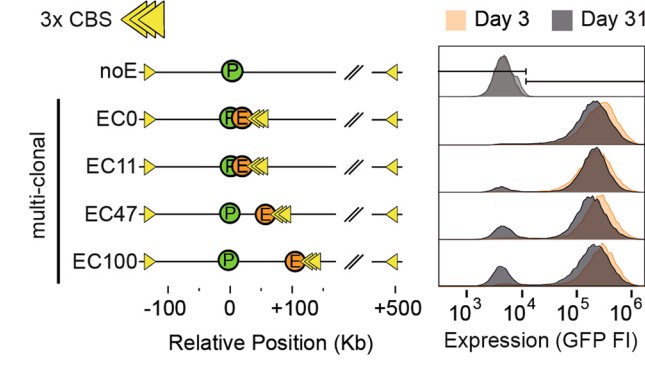

**Extended Data Fig. 1 | Distance-dependent enhancer protection against gene silencing in long-term cell cultures. a**. Overlay FACS profiles of multi-clonal cell populations without (noE) or with the enhancer integrated at a given distance (E0-E11-E47-E100kb) from the reporter gene, cultured for respectively 3 and 31 days after sorting for GFP-positive cells. The profiles show that over time, the gene silences with a rate that is related to enhancer distance. **b**. Transgene silencing coincides with H3K27me3 accumulation on transgene. ChIP-qPCR results showing relative enrichment of H3K27me3 at the integrated GFP reporter gene (Primer pair #1) or integrated HBG1-promoter (Primer pair #2) in E100 expressing GFP, E100 not expressing GFP and the noE cell line. Enrichment values on the y-axis are relative to that measured at a strong H3K27me3 site in K562 cells (see methods). For every condition two genomic positioned were assessed. Data are presented as mean values ± SD. For E100 GFP+, n = 3 and 2 biological replicates; For E100 GFP-, n = 2 and 2 biological replicates; For noE GFP-, n = 1 and 1 replicates. **c**. RNA quantification by reverse transcriptase qPCR of GFP driven by no enhancer (noE) or an enhancer at 0 kb (E0), 11 kb (E11), 47 kb (E47), 100 kb (E100), 407 kb (E407), in cells treated with hemin (purple) or not treated with hemin (green). Data are presented as mean values ± SD. n = 2 biological replicates per clone. **d**. As in **a**, but for clonally selected lines without (noE) or with the enhancer integrated at a given distance (E0-E11-E47-E100-E407kb) from the reporter gene. **e**. As in **a**, but for multi-clonal cell populations without (noE) or with a co-integrated enhancer-CBS at a given distance (EC0-11-47-100).

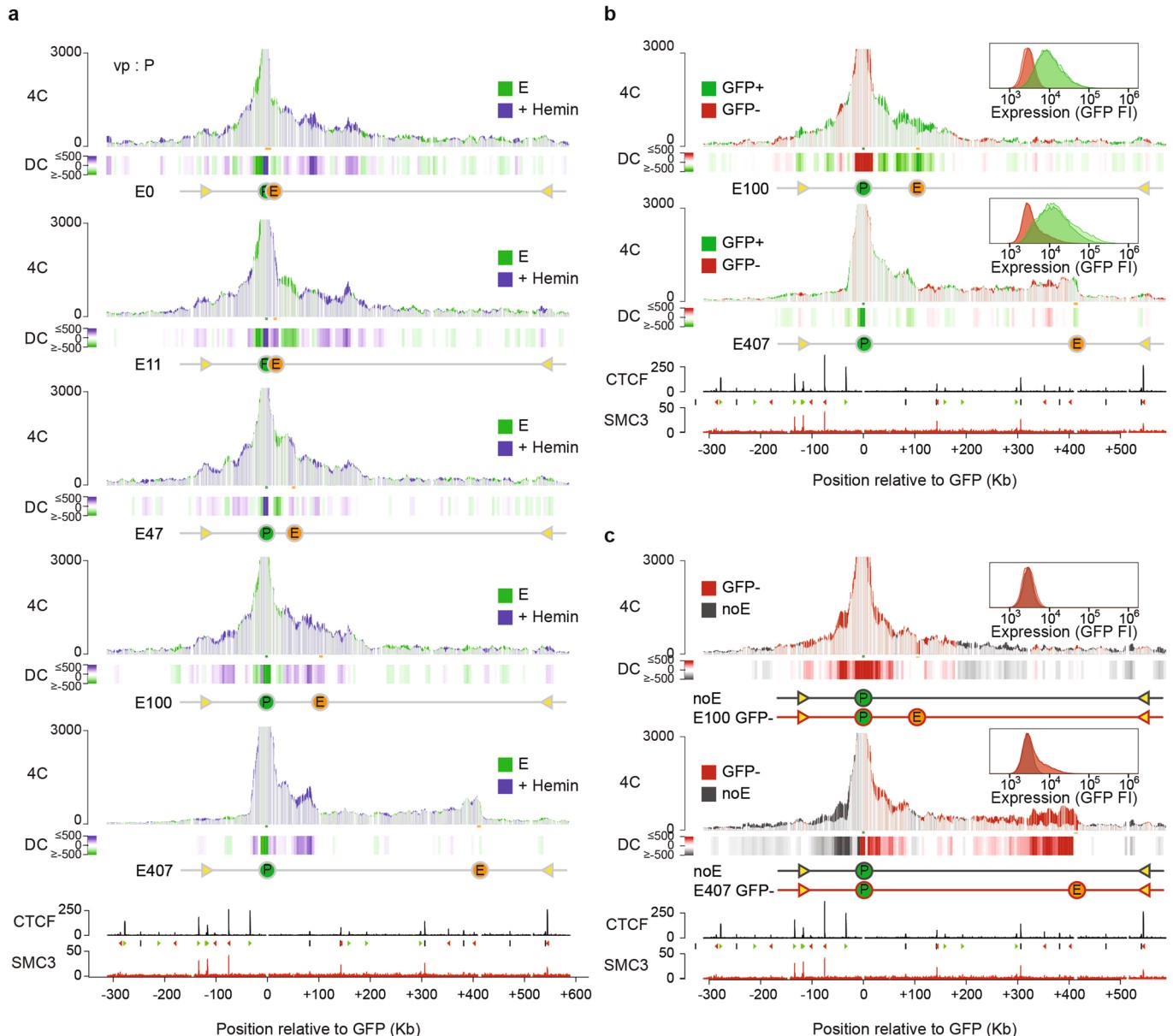

**Extended Data Fig. 2 | E-P contact frequencies change only mildly upon activation (hemin treatment) or silencing of the GFP reporter gene. a**. 4C-seq contact profile overlays comparing contacts of the integrated promoter (P) in the indicated E-lines (E0-11-47-100-407) cultured in the absence (green) or presence of Hemin (purple). Y-axis: 4C coverage normalized to 1 million *cis*-reads. Shared contacts are in light gray. Track below shows the differential contacts (DC) per 5 kb binned fragments. n= two technical replicates/clone. CTCF and SMC3 ChIP-seq signal tracks are shown for reference. Positions are in kb with respect to the integrated GFP reporter gene. **b**. 4C-seq contact profile overlays comparing the contacts of the integrated GFP reporter gene promoter in the expressing (green) and long-term silenced (red) E100 (top) and E407 (bottom) cell lines. **c**. 4C-seq contact profile overlays comparing the contacts of the integrated GFP reporter gene promoter in long-term silenced (red) E100 (top) and E407 (bottom) cell lines, versus its contacts in the noE cell line (lacking the enhancer). In **b,c** DC-tracks are plotted below each overlay. CTCF and SMC3 ChIP-seq signal tracks are shown for reference. Positions are in kb with respect to the integrated GFP reporter gene. GFP expression (FI) for the expressing (green), long-term silenced cell populations (red) or noE (darkgrey) are shown in the top right of each panel.

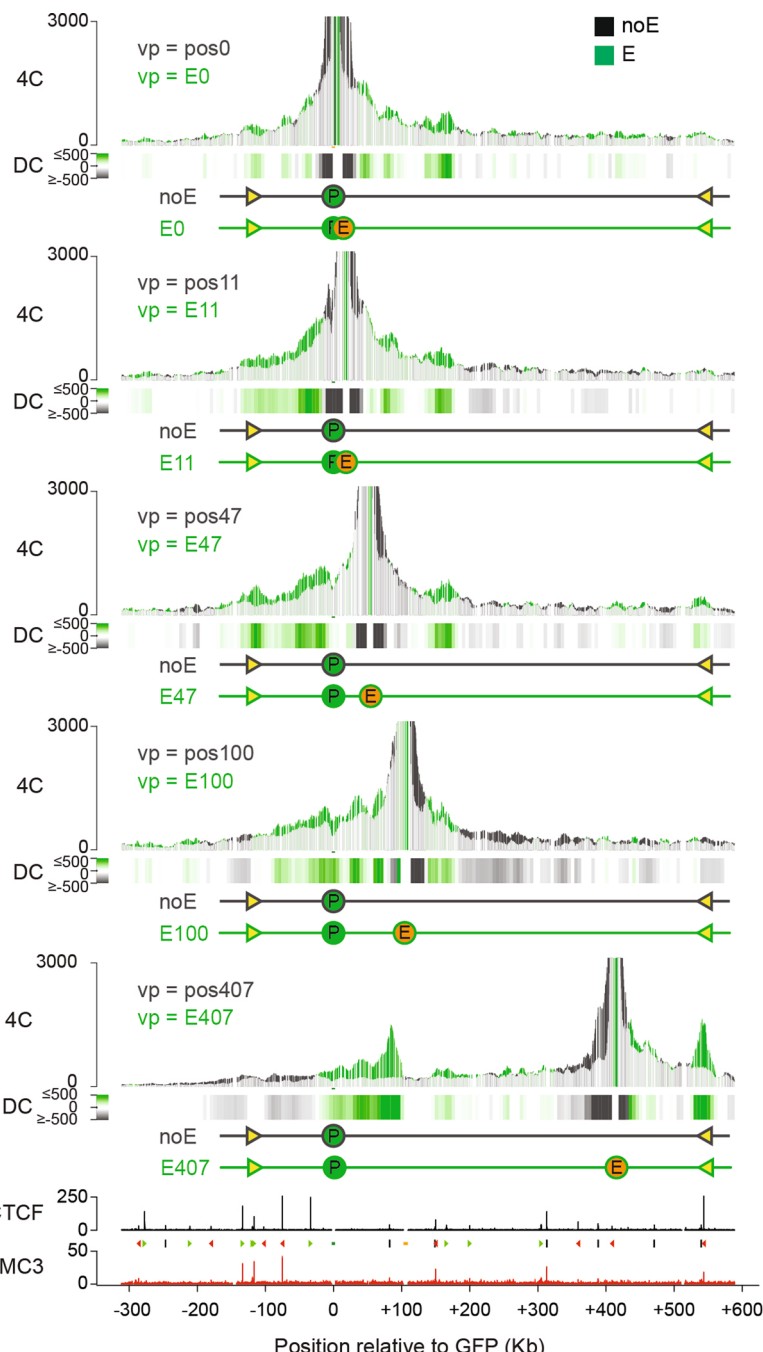

**Extended Data Fig. 3 | Tissue-specific enhancer stimulates the formation of local self-interacting chromatin domains.** The integrated enhancer engages in new contacts with surrounding sequences. 4C-seq contact profile overlays comparing contacts of the enhancer integrated at the indicated positions, versus contacts of the corresponding endogenous location in cells lacking an integrated enhancer (noE cells). Y-axis: 4C coverage normalized per 1 million *cis*-reads. Shared contacts are in light gray. Track below shows the differential contacts (DC) for fragments binned per 5 Kb. n= two technical replicates/clone.

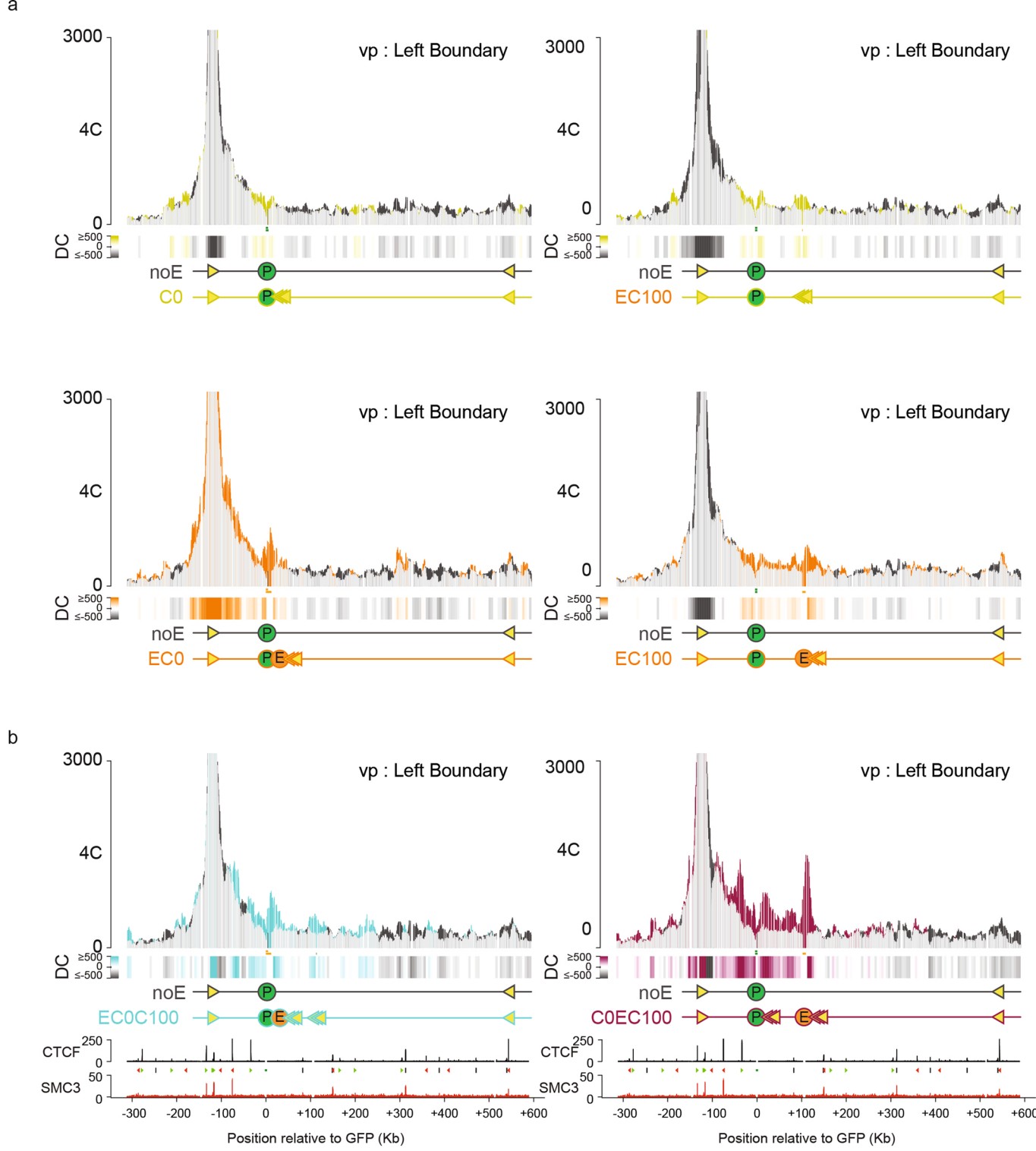

**Extended Data Fig. 4 | Enhancer selects flanking CTCF sites for engagement in chromatin looping. a**. 4C-seq contact profile overlays showing contacts gained by the left endogenous boundary in C lines (yellow, no enhancer, top) or EC lines (orange, with enhancer, bottom), as compared its contacts in the cell line lacking an integrated enhancer or integrated CTCF sites (noE: dark gray). Shared contacts are in light gray. y axis: 4C coverage normalized per 1 million *cis* = reads. n= two technical replicates/clone. **b**. 4C-seq profile overlays with the left plot showing in blue contacts gained by the left endogenous boundary in the cell line having the enhancer at E0 and two CTCF cassettes at C0 and C100, and the right plot showing in red contacts gained by the left endogenous boundary in the cell line having the same two CTCF cassettes at C0 and C100, but the enhancer at E100. DC-tracks plotted below each overlay. n= two technical replicates/clone.

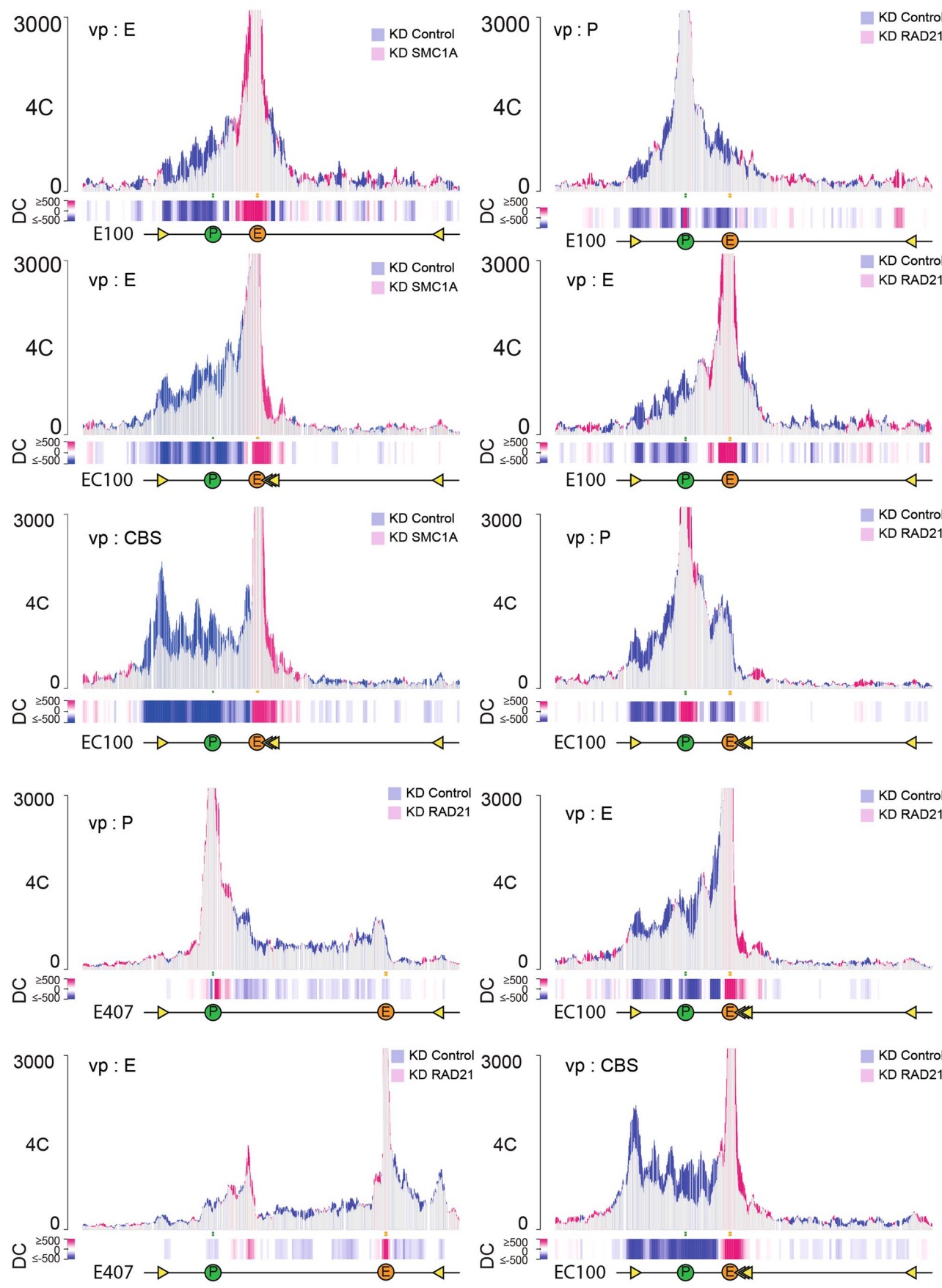

**Extended Data Fig. 5 | See next page for caption.**

**Extended Data Fig. 5 | Tissue-specific enhancer relies on cohesin to create self-interacting domains.** 4C-seq contact profile overlays comparing contacts of the integrated GFP reporter gene promoter (P), μLCR enhancer (E) and 3x-hCTCF (CBS), in the indicated (E100, EC100 and E407) cell lines treated with a control sgRNA (blue) versus a RAD21 or SMC1A sgRNA (causing partial depletion, in pink). Shared contacts are in light gray. Y-axis: 4C coverage normalized to 1 million *cis*-reads.

|---|---|

# Reporting Summary

## Statistics

For all statistical analyses, confirm that the following items are present in the figure legend, table legend, main text, or Methods section.

| n/a | Confirmed | |
|---|---|---|
| ☐ | ☒ | The exact sample size (*n*) for each experimental group/condition, given as a discrete number and unit of measurement |
| ☐ | ☒ | A statement on whether measurements were taken from distinct samples or whether the same sample was measured repeatedly |
| ☒ | ☐ | The statistical test(s) used AND whether they are one- or two-sided<br>*Only common tests should be described solely by name; describe more complex techniques in the Methods section.* |
| ☒ | ☐ | A description of all covariates tested |
| ☒ | ☐ | A description of any assumptions or corrections, such as tests of normality and adjustment for multiple comparisons |
| ☐ | ☒ | A full description of the statistical parameters including central tendency (e.g. means) or other basic estimates (e.g. regression coefficient) AND variation (e.g. standard deviation) or associated estimates of uncertainty (e.g. confidence intervals) |
| ☒ | ☐ | For null hypothesis testing, the test statistic (e.g. *F*, *t*, *r*) with confidence intervals, effect sizes, degrees of freedom and *P* value noted<br>*Give P values as exact values whenever suitable.* |
| ☒ | ☐ | For Bayesian analysis, information on the choice of priors and Markov chain Monte Carlo settings |
| ☒ | ☐ | For hierarchical and complex designs, identification of the appropriate level for tests and full reporting of outcomes |
| ☒ | ☐ | Estimates of effect sizes (e.g. Cohen's *d*, Pearson's *r*), indicating how they were calculated |

*Our web collection on statistics for biologists contains articles on many of the points above.*

## Software and code

Policy information about availability of computer code

| Data collection | CytExpert 2.3.0.84, Bio-Rad CFX Manager 2.1 |
|---|---|
| Data analysis | Plots were generated in R (version 4.1.0), FlowJo 10.5.2 and GraphPad Prism 8.2.1. Sequencing was analyzed with pipeline "pipe4C" (https://github.com/deLaatLab/pipe4C) and custom R base code, as indicated in text. |

For manuscripts utilizing custom algorithms or software that are central to the research but not yet described in published literature, software must be made available to editors and reviewers. We strongly encourage code deposition in a community repository (e.g. GitHub). See the Nature Portfolio guidelines for submitting code & software for further information.

## Data

Policy information about availability of data

All manuscripts must include a data availability statement. This statement should provide the following information, where applicable:
- Accession codes, unique identifiers, or web links for publicly available datasets
- A description of any restrictions on data availability
- For clinical datasets or third party data, please ensure that the statement adheres to our policy

Public ChIP-seq and RNA-seq (mapped to human genome release 19) tracks were downloaded from ENCODE portal9. The following data-sets from ENCODE were used: K562 SMC3 ChIP-seq (Encode, Michael Snyder, ENCSR000EGW, ENCFF479BWQ), ChIP-seq CTCF ChIP-seq (Encode, Bradley Bernstein, ENCSR000AKO, ENCFF000BWF), K562 RNA-seq (Encode, Brenton Graveley, ENCSR000AEN, ENCFF657EOD, ENCFF578WIM). H3K27ac (Encode, Bradley Bernstein, GEO: GSM733656) and H3K27me3 (Encode, Michael Snyder, GEO: GSM788088).

Raw sequencing data and mapped wig files of the following experiments are available without restriction from the Gene Expression Omnibus (GEO) under accession

March 2021

GSE180566:

Sample Technique Genotype Viewpoint Hemin Sorted KnockDown Genome Figure
Sample_1 4C WT Pos0 no_Hemin GFP_negative NA hg19_with_insert 2a
Sample_2 4C WT Pos0 no_Hemin GFP_negative NA hg19_with_insert 2a
Sample_3 4C noE Pos0 no_Hemin GFP_negative NA hg19_with_insert 2c, ext. 3
Sample_4 4C noE Pos0 no_Hemin GFP_negative NA hg19_with_insert 2c, ext. 3
Sample_5 4C noE GFP no_Hemin GFP_negative NA hg19_with_insert 2a, 2b, 2c, 2e, 4b, 4c, 4j, ext. 2c
Sample_6 4C noE GFP no_Hemin GFP_negative NA hg19_with_insert 2a, 2b, 2c, 2e, 4b, 4c, 4j, ext. 2c
Sample_7 4C noE GFP no_Hemin GFP_negative NA hg19_with_insert 2a, 2b, 2c, 2e, 4b, 4c, 4j, ext. 2c
Sample_8 4C E0 GFP no_Hemin 2xGFP_positive NA hg19_with_insert 1j, 2b, 2c, 3c, ext. 2a
Sample_9 4C E0 GFP no_Hemin 2xGFP_positive NA hg19_with_insert 1j, 2b, 2c, 3c, ext. 2a
Sample_10 4C E11 GFP no_Hemin 2xGFP_positive NA hg19_with_insert 1j, 2b, 2c, ext. 2a
Sample_11 4C E11 GFP no_Hemin 2xGFP_positive NA hg19_with_insert 1j, 2b, 2c, ext. 2a
Sample_12 4C E47 GFP no_Hemin 2xGFP_positive NA hg19_with_insert 1j, 2b, 2c, ext. 2a
Sample_13 4C E47 GFP no_Hemin 2xGFP_positive NA hg19_with_insert 1j, 2b, 2c, ext. 2a
Sample_14 4C E100 GFP no_Hemin 2xGFP_positive NA hg19_with_insert 1j, 2b, 2c, 3c, 3e, 4b, ext. 2a
Sample_15 4C E100 GFP no_Hemin 2xGFP_positive NA hg19_with_insert 1j, 2b, 2c, 3c, 3e, 4b, ext. 2a
Sample_16 4C E407 GFP no_Hemin 2xGFP_positive NA hg19_with_insert 1j, 2e, 2d, ext. 2a
Sample_17 4C E407 GFP no_Hemin 2xGFP_positive NA hg19_with_insert 1j, 2e, 2d, ext. 2a
Sample_18 4C E407 GFP no_Hemin 2xGFP_positive NA hg19_with_insert 1j, 2e, 2d, ext. 2a
Sample_19 4C EC0 GFP no_Hemin 2xGFP_positive NA hg19_with_insert 3c, 3d
Sample_20 4C EC0 GFP no_Hemin 2xGFP_positive NA hg19_with_insert 3c, 3d
Sample_21 4C EC0 GFP no_Hemin 2xGFP_positive NA hg19_with_insert 3c, 3d
Sample_22 4C E0 uLCR no_Hemin 2xGFP_positive NA hg19_with_insert 2c, 3c, ext. 3
Sample_23 4C E0 uLCR no_Hemin 2xGFP_positive NA hg19_with_insert 2c, 3c, ext. 3
Sample_24 4C EC0 uLCR no_Hemin 2xGFP_positive NA hg19_with_insert 3c
Sample_25 4C EC0 uLCR no_Hemin 2xGFP_positive NA hg19_with_insert 3c
Sample_26 4C EC0 uLCR no_Hemin 2xGFP_positive NA hg19_with_insert 3c
Sample_27 4C EC100 GFP no_Hemin 2xGFP_positive NA hg19_with_insert 3c, 3d, 4c
Sample_28 4C EC100 GFP no_Hemin 2xGFP_positive NA hg19_with_insert 3c, 3d, 4c
Sample_29 4C E100 uLCR no_Hemin 2xGFP_positive NA hg19_with_insert 2c, 3c, 3e, 4b, ext. 3
Sample_30 4C E100 uLCR no_Hemin 2xGFP_positive NA hg19_with_insert 2c, 3c, 3e, 4b, ext. 3
Sample_31 4C EC100 uLCR no_Hemin 2xGFP_positive NA hg19_with_insert 3c, 4c
Sample_32 4C EC100 uLCR no_Hemin 2xGFP_positive NA hg19_with_insert 3c, 4c
Sample_33 4C EC100 uLCR no_Hemin 2xGFP_positive NA hg19_with_insert 3c, 4c
Sample_34 4C C0 GFP no_Hemin GFP_negative NA hg19_with_insert 3d
Sample_35 4C C0 GFP no_Hemin GFP_negative NA hg19_with_insert 3d
Sample_36 4C C0 GFP no_Hemin GFP_negative NA hg19_with_insert 3d
Sample_37 4C C0 CBS no_Hemin GFP_negative NA hg19_with_insert 3d
Sample_38 4C C0 CBS no_Hemin GFP_negative NA hg19_with_insert 3d
Sample_39 4C EC0 CBS no_Hemin 2xGFP_positive NA hg19_with_insert 3d
Sample_40 4C EC0 CBS no_Hemin 2xGFP_positive NA hg19_with_insert 3d
Sample_41 4C C100 GFP no_Hemin GFP_negative NA hg19_with_insert 3d
Sample_42 4C C100 GFP no_Hemin GFP_negative NA hg19_with_insert 3d
Sample_43 4C C100 CBS no_Hemin GFP_negative NA hg19_with_insert 3d
Sample_44 4C C100 CBS no_Hemin GFP_negative NA hg19_with_insert 3d
Sample_45 4C EC100 CBS no_Hemin 2xGFP_positive NA hg19_with_insert 3d, 4c
Sample_46 4C EC100 CBS no_Hemin 2xGFP_positive NA hg19_with_insert 3d, 4c
Sample_47 4C C-0E100 GFP no_Hemin 2xGFP_positive NA hg19_with_insert 3e
Sample_48 4C C-0E100 GFP no_Hemin 2xGFP_positive NA hg19_with_insert 3e
Sample_49 4C C-0E100 GFP no_Hemin 2xGFP_positive NA hg19_with_insert 3e
Sample_50 4C C-0E100 uLCR no_Hemin 2xGFP_positive NA hg19_with_insert 3e
Sample_51 4C C-0E100 uLCR no_Hemin 2xGFP_positive NA hg19_with_insert 3e
Sample_52 4C C-0E100 uLCR no_Hemin 2xGFP_positive NA hg19_with_insert 3e
Sample_53 4C E100 GFP no_Hemin 2xGFP_positive Control_KD hg19_with_insert 4a
Sample_54 4C E100 GFP no_Hemin 2xGFP_positive SMC1A_KD hg19_with_insert 4a, 4b
Sample_55 4C EC100 GFP no_Hemin 2xGFP_positive Control_KD hg19_with_insert 4a
Sample_56 4C EC100 GFP no_Hemin 2xGFP_positive Control_KD hg19_with_insert 4a
Sample_57 4C EC100 GFP no_Hemin 2xGFP_positive SMC1A_KD hg19_with_insert 4a, 4c
Sample_58 4C E100 GFP no_Hemin 2xGFP_positive RAD21_KD hg19_with_insert 4b, ext. 5
Sample_59 4C noE Pos100 no_Hemin GFP_negative NA hg19_with_insert 2c, 4b, 4c, ext. 3
Sample_60 4C noE Pos100 no_Hemin GFP_negative NA hg19_with_insert 2c, 4b, 4c, ext. 3
Sample_61 4C E100 uLCR no_Hemin 2xGFP_positive SMC1A_KD hg19_with_insert 4b, ext. 5
Sample_62 4C E100 uLCR no_Hemin 2xGFP_positive RAD21_KD hg19_with_insert 4b, ext. 5
Sample_63 4C EC100 GFP no_Hemin 2xGFP_positive RAD21_KD hg19_with_insert 4c, ext. 5
Sample_64 4C EC100 uLCR no_Hemin 2xGFP_positive SMC1A_KD hg19_with_insert 4c, ext. 5
Sample_65 4C EC100 uLCR no_Hemin 2xGFP_positive RAD21_KD hg19_with_insert 4c, ext. 5
Sample_66 4C EC100 CBS no_Hemin 2xGFP_positive SMC1A_KD hg19_with_insert 4c, ext. 5
Sample_67 4C EC100 CBS no_Hemin 2xGFP_positive RAD21_KD hg19_with_insert 4c, ext. 5
Sample_68 4C E407 GFP no_Hemin 2xGFP_positive Control_KD_ hg19_with_insert 4d, ext. 5
Sample_69 4C E407 GFP no_Hemin 2xGFP_positive Control_KD_ hg19_with_insert 4d, ext. 5
Sample_70 4C E407 GFP no_Hemin 2xGFP_positive Control_KD_ hg19_with_insert 4d, ext. 5
Sample_71 4C E407 GFP no_Hemin 2xGFP_positive Control_SMC1A_ hg19_with_insert 4d
Sample_72 4C E407 uLCR no_Hemin 2xGFP_positive Control_KD_ hg19_with_insert 4d, ext. 5
Sample_73 4C E407 uLCR no_Hemin 2xGFP_positive Control_KD_ hg19_with_insert 4d, ext. 5

Sample_74 4C E407 uLCR no_Hemin 2xGFP_positive SMC1A_KD_ hg19_with_insert 4d
Sample_75 4C E0C0C100 GFP no_Hemin 2xGFP_positive NA hg19_with_insert 4j
Sample_76 4C E0C0C100 GFP no_Hemin 2xGFP_positive NA hg19_with_insert 4j
Sample_77 4C E100C0C100 GFP no_Hemin 2xGFP_positive NA hg19_with_insert 4j
Sample_78 4C E100C0C100 GFP no_Hemin 2xGFP_positive NA hg19_with_insert 4j
Sample_79 4C E0 GFP plus_Hemin 2xGFP_positive NA hg19_with_insert 1j, ext. 2a
Sample_80 4C E0 GFP plus_Hemin 2xGFP_positive NA hg19_with_insert 1j, ext. 2a
Sample_81 4C E11 GFP plus_Hemin 2xGFP_positive NA hg19_with_insert 1j, ext. 2a
Sample_82 4C E11 GFP plus_Hemin 2xGFP_positive NA hg19_with_insert 1j, ext. 2a
Sample_83 4C E47 GFP plus_Hemin 2xGFP_positive NA hg19_with_insert 1j, ext. 2a
Sample_84 4C E47 GFP plus_Hemin 2xGFP_positive NA hg19_with_insert 1j, ext. 2a
Sample_85 4C E100 GFP plus_Hemin 2xGFP_positive NA hg19_with_insert 1j, ext. 2a
Sample_86 4C E100 GFP plus_Hemin 2xGFP_positive NA hg19_with_insert 1j, ext. 2a
Sample_87 4C E100 GFP plus_Hemin 2xGFP_positive NA hg19_with_insert 1j, ext. 2a
Sample_88 4C E407 GFP plus_Hemin 2xGFP_positive NA hg19_with_insert 1j, ext. 2a
Sample_89 4C E407 GFP plus_Hemin 2xGFP_positive NA hg19_with_insert 1j, ext. 2a
Sample_90 4C E407 GFP plus_Hemin 2xGFP_positive NA hg19_with_insert 1j, ext. 2a
Sample_91 4C E100 GFP no_Hemin 6xGFP_negative NA hg19_with_insert 1j
Sample_92 4C E100 GFP no_Hemin 6xGFP_negative NA hg19_with_insert 1j
Sample_96 4C E407 GFP no_Hemin 9xGFP_negative NA hg19_with_insert 1j
Sample_97 4C E407 GFP no_Hemin 9xGFP_negative NA hg19_with_insert 1j
Sample_98 4C E407 GFP no_Hemin 9xGFP_negative NA hg19_with_insert 1j
Sample_101 4C noE Pos11 no_Hemin GFP_negative NA hg19_with_insert 2c, ext. 3
Sample_102 4C noE Pos11 no_Hemin GFP_negative NA hg19_with_insert 2c, ext. 3
Sample_105 4C noE Pos47 no_Hemin GFP_negative NA hg19_with_insert 2c, ext. 3
Sample_106 4C noE Pos47 no_Hemin GFP_negative NA hg19_with_insert 2c, ext. 3
Sample_111 4C noE Pos407 no_Hemin GFP_negative NA hg19_with_insert 2d, ext. 3
Sample_112 4C noE Pos407 no_Hemin GFP_negative NA hg19_with_insert 2d, ext. 3
Sample_113 4C E11 uLCR no_Hemin 2xGFP_positive NA hg19_with_insert 2c, ext. 3
Sample_114 4C E11 uLCR no_Hemin 2xGFP_positive NA hg19_with_insert 2c, ext. 3
Sample_115 4C E47 uLCR no_Hemin 2xGFP_positive NA hg19_with_insert 2c, ext. 3
Sample_116 4C E47 uLCR no_Hemin 2xGFP_positive NA hg19_with_insert 2c, ext. 3
Sample_117 4C E407 uLCR no_Hemin 2xGFP_positive NA hg19_with_insert 2d, ext. 3
Sample_118 4C E407 uLCR no_Hemin 2xGFP_positive NA hg19_with_insert 2d, ext. 3
Sample_119 4C E407 uLCR no_Hemin 2xGFP_positive NA hg19_with_insert 2d, ext. 3
Sample_120 4C noE LeftBoundary no_Hemin GFP_negative NA hg19_with_insert ext. 4
Sample_121 4C noE LeftBoundary no_Hemin GFP_negative NA hg19_with_insert ext. 4
Sample_122 4C EC0 LeftBoundary no_Hemin 2xGFP_positive NA hg19_with_insert ext. 4
Sample_123 4C EC0 LeftBoundary no_Hemin 2xGFP_positive NA hg19_with_insert ext. 4
Sample_124 4C EC100 LeftBoundary no_Hemin 2xGFP_positive NA hg19_with_insert ext. 4
Sample_125 4C EC100 LeftBoundary no_Hemin 2xGFP_positive NA hg19_with_insert ext. 4
Sample_126 4C E0C0C100 LeftBoundary no_Hemin 2xGFP_positive NA hg19_with_insert ext. 4
Sample_127 4C E0C0C100 LeftBoundary no_Hemin 2xGFP_positive NA hg19_with_insert ext. 4
Sample_128 4C E100C0C100 LeftBoundary no_Hemin 2xGFP_positive NA hg19_with_insert ext. 4
Sample_129 4C E100C0C100 LeftBoundary no_Hemin 2xGFP_positive NA hg19_with_insert ext. 4
Sample_130 4C EC100 uLCR no_Hemin 2xGFP_positive Control_KD hg19_with_insert ext. 5
Sample_131 4C EC100 GFP no_Hemin 2xGFP_positive Control_KD hg19_with_insert ext. 5
Sample_132 4C EC100 GFP no_Hemin 2xGFP_positive Control_KD hg19_with_insert ext. 5
Sample_133 4C EC100 uLCR no_Hemin 2xGFP_positive Control_KD hg19_with_insert ext. 5
Sample_134 4C EC100 uLCR no_Hemin 2xGFP_positive Control_KD hg19_with_insert ext. 5
Sample_135 4C EC100 CBS no_Hemin 2xGFP_positive Control_KD hg19_with_insert ext. 5
Sample_136 4C EC100 CBS no_Hemin 2xGFP_positive Control_KD hg19_with_insert ext. 5
Sample_137 4C E407 GFP no_Hemin 2xGFP_positive RAD21_KD hg19_with_insert ext. 5
Sample_138 4C E407 uLCR no_Hemin 2xGFP_positive RAD21_KD hg19_with_insert ext. 5
Sample_139 4C EC100 CBS no_Hemin 2xGFP_positive Control_KD hg19_with_insert ext. 5
Sample_140 4C EC100 CBS no_Hemin 2xGFP_positive Control_KD hg19_with_insert ext. 5
Sample_141 4C E47 GFP no_Hemin 2xGFP_positive NA hg19_with_insert 1j
Sample_142 4C E47 GFP no_Hemin 2xGFP_positive NA hg19_with_insert 1j
Sample_143 4C E100 GFP no_Hemin 2xGFP_positive NA hg19_with_insert 1j, ext. 2b
Sample_144 4C E100 GFP no_Hemin 2xGFP_positive NA hg19_with_insert 1j, ext. 2b
Sample_145 4C E407 GFP no_Hemin 2xGFP_positive NA hg19_with_insert 1j, ext. 2b
Sample_146 4C E407 GFP no_Hemin 2xGFP_positive NA hg19_with_insert 1j, ext. 2b
Sample_147 4C E100 GFP no_Hemin 6xGFP_negative NA hg19_with_insert 1j, ext. 2b, 2c
Sample_148 4C E100 GFP no_Hemin 6xGFP_negative NA hg19_with_insert 1j, ext. 2b, 2c
Sample_149 4C E407 GFP no_Hemin 9xGFP_negative NA hg19_with_insert 1j, ext. 2b, 2c
Sample_150 4C E407 GFP no_Hemin 9xGFP_negative NA hg19_with_insert 1j, ext. 2b, 2c
Sample_151 4C E407 GFP no_Hemin 9xGFP_negative NA hg19_with_insert 1j
Sample_152 4C E407 GFP no_Hemin 9xGFP_negative NA hg19_with_insert 1j
Sample_153 4C C0 LeftBoundary no_Hemin GFP_negative NA hg19_with_insert ext. 4
Sample_154 4C C0 LeftBoundary no_Hemin GFP_negative NA hg19_with_insert ext. 4
Sample_155 4C C0 LeftBoundary no_Hemin GFP_negative NA hg19_with_insert ext. 4
Sample_156 4C C100 LeftBoundary no_Hemin GFP_negative NA hg19_with_insert ext. 4
Sample_157 4C C100 LeftBoundary no_Hemin GFP_negative NA hg19_with_insert ext. 4

# Field-specific reporting

Please select the one below that is the best fit for your research. If you are not sure, read the appropriate sections before making your selection.

☒ Life sciences        ☐ Behavioural & social sciences        ☐ Ecological, evolutionary & environmental sciences

For a reference copy of the document with all sections, see nature.com/documents/nr-reporting-summary-flat.pdf

# Life sciences study design

All studies must disclose on these points even when the disclosure is negative.

| | |
|---|---|
| Sample size | No statistical method was used to predetermine sample size |
| Data exclusions | no data were excluded from the analyses |
| Replication | Number of replicates are indicated in the figure legends. All 4C and all but one ChIP-qPCR experiments represent at least two technical replicates.<br>Additionally, all conclusions are based on the analysis of multiple independent clones having the enhancer at different distances and/or having ectopic CTCF sites at different positions and/or having different cohesin subunits depleted. |
| Randomization | the experiments were not randomized |
| Blinding | The investigators were not blinded to allocation during experiments and outcome assessment. |

# Reporting for specific materials, systems and methods

We require information from authors about some types of materials, experimental systems and methods used in many studies. Here, indicate whether each material, system or method listed is relevant to your study. If you are not sure if a list item applies to your research, read the appropriate section before selecting a response.

## Materials & experimental systems

| n/a | Involved in the study |
|---|---|
| ☐ | ☒ Antibodies |
| ☐ | ☒ Eukaryotic cell lines |
| ☒ | ☐ Palaeontology and archaeology |
| ☒ | ☐ Animals and other organisms |
| ☒ | ☐ Human research participants |
| ☒ | ☐ Clinical data |
| ☒ | ☐ Dual use research of concern |

## Methods

| n/a | Involved in the study |
|---|---|
| ☒ | ☐ ChIP-seq |
| ☐ | ☒ Flow cytometry |
| ☒ | ☐ MRI-based neuroimaging |

## Antibodies

| | |
|---|---|
| Antibodies used | SMC1: A300-055A, for western 1:1000, for ChIP 5ug per 30ug of chromatin, Bethyl, https://www.thermofisher.com/antibody/product/SMC1-Antibody-Polyclonal/A300-055A<br>CTCF: 07-829, ChIP 5ug per 30ug of chromatin, Millipore, https://www.merckmillipore.com/NL/en/product/Anti-CTCF-Antibody,MM_NF-07-729?ReferrerURL=https%3A%2F%2Fwww.google.com%2F<br>H3K27me3: ab6002, ChIP 5ug per 30ug of chromatin, Abcam, https://www.abcam.com/Histone-H3-tri-methyl-K27-antibody-mAbcam-6002-ChIP-Grade-ab6002.html?gclsrc=aw.ds\|aw.ds&gclid=CjwKCAjwxOCRBhA8EiwA0X8hi4XuFW8mffpMVkwU6GjRPPeanemAtd_AXf5jc85V5YQ2EJR1ftOxJhoCL9oQAvD_BwE<br>y-Tubulin: GTU-88 T6557, 1:3000, Sigma, https://www.sigmaaldrich.com/NL/en/product/sigma/t6557<br>goat anti rabbit HRP: #7074, 1:3000, Cell Signalling, https://www.cellsignal.com/products/secondary-antibodies/anti-rabbit-igg-hrp-linked-antibody/7074<br>goat anti mouse HRP: #7076, 1:3000, Cell, Signalling, https://www.cellsignal.com/products/secondary-antibodies/anti-mouse-igg-hrp-linked-antibody/7076 |
| Validation | Antibody validation can be found on manufacturers website as indicated. |

# Eukaryotic cell lines

Policy information about cell lines

| | |
|---|---|
| Cell line source(s) | Human erythroleukemia K562 cells were used in this study (not authenticated cell line, available in our institute, periodically tested for mycoplasma). |
| Authentication | K562 cells were not authenticated. |
| Mycoplasma contamination | K562 cells were regularly testes for mycoplasma. |
| Commonly misidentified lines (See ICLAC register) | No commonly misidentified cell lines were used. |

# Flow Cytometry

## Plots

Confirm that:

☒ The axis labels state the marker and fluorochrome used (e.g. CD4-FITC).

☒ The axis scales are clearly visible. Include numbers along axes only for bottom left plot of group (a 'group' is an analysis of identical markers).

☒ All plots are contour plots with outliers or pseudocolor plots.

☒ A numerical value for number of cells or percentage (with statistics) is provided.

## Methodology

| | |
|---|---|
| Sample preparation | For flow cytometry analysis, genetically modified K562 cells were cultured as indicated in methods section and either treated with or without hemin two days prior to analysis. Cells were transferred to a 96-wells plate for flow cytometry analysis directly from the culture dish. For every experiment, at least 10,000 cells within the live single-cell gate were recorded. For fluorescence assisted cell sorting, 10 million cells were collected, centrifuged and resuspended in 1 mL of culture medium. |
| Instrument | Flow cytometry analysis: Cytoflex S, Beckman Coulter Inc. Model No.: B75442<br>Fluorescence assisted cell sorting: FACS Aria Fusion, Beckman Coulter Inc. Special Order Research Product |
| Software | Flow cytometry analysis: CytExpert 2.3.0.84<br>Fluorescence assisted cell sorting: FACSDiva 8.0.1, Beckman Coulter Inc. |
| Cell population abundance | We used identical gate settings to separate GFP (or dsRed) positive and negative cells for all experiments, as described in the manuscript. |
| Gating strategy | For flow cytometry analysis and fluorescence assisted cell sorting, cells were gated for live single cells based on FSC-A, SSC-A and FSC-H.<br>For sorting, cells were considered GFP positive based on gating on "noE" cell line. Gates were drawn such that more than 99.9% of "noE" cells felt into the GFP-negative gate.<br>To enrich for KRAB::BFP containing cells prior to knock-down experiments, cells were gated for BFP positive. From this population, the top 50% BFP expressing cells were sorted.<br>For analysis of knock-down experiments, cells were gated to be BFP/sgRNA positive as compared to cells that only contained KRAB::BFP. |

☒ Tick this box to confirm that a figure exemplifying the gating strategy is provided in the Supplementary Information.

