## [Peer Review File. · Nature Structural & Molecular Biology]

Peer Review Information

Journal: Nature Structural and Molecular Biology

Manuscript Title: Building regulatory landscapes reveals that an enhancer can recruit cohesin to create contact domains, engage CTCF sites and activate distant genes

Corresponding author name(s): Professor Wouter de Laat

Editorial Notes:

Redactions – transferred manuscripts (mention of previous referee reports from elsewhere) This manuscript has been previously reviewed at another journal. This document only contains reviewer comments, rebuttal and decision letters for versions considered at Nature **NSMB**. Mentions of prior referee reports have been redacted

Reviewer Comments & Decisions:

Decision Letter, initial version:

7th Mar 2022

Dear Dr. de Laat,

Thank you for submitting your revised manuscript "Building regulatory landscapes: enhancers recruit cohesin to create contact domains, engage CTCF sites and activate distant genes" (NSMB-A45822-T). It has now been seen by the original referees and their comments are below. The reviewers find that the paper has improved in revision, and therefore we'll be happy in principle to publish it in Nature Structural & Molecular Biology, pending minor revisions to satisfy the referees' final requests and to comply with our editorial and formatting guidelines.

Sincerely,

Carolina

Carolina Perdigoto, PhD
Chief Editor
Nature Structural & Molecular Biology
orcid.org/0000-0002-5783-7106

Reviewer #1 (Remarks to the Author):

The authors have addressed all my concerns and improved their manuscript with additional data and better explanations. I would be happy to recommend publication in NSMB.

Reviewer #2 (Remarks to the Author):

The authors provided a very thoughtful response to the reviewers. I have no more comments. The manuscript presents a very strong and impactful body of work and i recommend publication without any further delay.

Reviewer #3 (Remarks to the Author):

I am satisfied with the revisions. It's a bit of a shame that Point 6 about the enhancer cannot be addressed straightforwardly due to the experimental design, but the added explanation addresses this adequately.

Final Decision Letter:

25th Apr 2022

Dear Dr. de Laat,

We are now happy to accept your revised paper "Building regulatory landscapes reveals that an enhancer can recruit cohesin to create contact domains, engage CTCF sites and activate distant genes" for publication as a Article in Nature Structural & Molecular Biology.

As soon as your article is published, you can generate your shareable link by entering the DOI of your article here: http://authors.springernature.com/share.

Corresponding authors will also receive an automated email with the shareable link

Note the policy of the journal on data deposition:

<http://www.nature.com/authors/policies/availability.html>.

Your paper will be published online soon after we receive proof corrections and will appear in print in the next available issue. You can find out your date of online publication by contacting the production team shortly after sending your proof corrections. Content is published online weekly on Mondays and Thursdays, and the embargo is set at 16:00 London time (GMT)/11:00 am US Eastern time (EST) on the day of publication. Now is the time to inform your Public Relations or Press Office about your paper, as they might be interested in promoting its publication. This will allow them time to prepare an accurate and satisfactory press release. Include your manuscript tracking number (NSMB-A45822A) and our journal name, which they will need when they contact our press office.

About one week before your paper is published online, we shall be distributing a press release to news organizations worldwide, which may very well include details of your work. We are happy for your institution or funding agency to prepare its own press release, but it must mention the embargo date and Nature Structural & Molecular Biology. If you or your Press Office have any enquiries in the meantime, please contact press@nature.com.

Please note that *Nature Structural & Molecular Biology* is a Transformative Journal (TJ). Authors

may publish their research with us through the traditional subscription access route or make their paper immediately open access through payment of an article-processing charge (APC). Authors will not be required to make a final decision about access to their article until it has been accepted. [Find out more about Transformative Journals](https://www.springernature.com/gp/open-research/transformative-journals)

Authors may need to take specific actions to achieve [compliance with funder and institutional open access mandates](https://www.springernature.com/gp/open-research/funding/policy-compliance-faqs). If your research is supported by a funder that requires immediate open access (e.g. according to [Plan S principles](https://www.springernature.com/gp/open-research/plan-s-compliance)) then you should select the gold OA route, and we will direct you to the compliant route where possible. For authors selecting the subscription publication route, the journal's standard licensing terms will need to be accepted, including [self-archiving policies](https://www.springernature.com/gp/open-research/policies/journal-policies). Those licensing terms will supersede any other terms that the author or any third party may assert apply to any version of the manuscript.

Sincerely,

Carolina Perdigoto, PhD
Chief Editor
Nature Structural & Molecular Biology
orcid.org/0000-0002-5783-7106